# In-Line Holographic Droplet Imaging: Accelerated Classification with Convolutional Neural Networks and Quantitative Experimental Validation

Birte Thiede<sup>1,2</sup>, Oliver Schlenczek<sup>1</sup>, Katja Stieger<sup>1,2</sup>, Alexander Ecker<sup>1,3</sup>, Eberhard Bodenschatz<sup>1,2,4</sup>, and Gholamhossein Bagheri<sup>1</sup>

**Correspondence:** Gholamhossein Bagheri (gholamhossein.bagheri@ds.mpg.de)

Abstract. Accurate measurements of cloud particle size, shape, and concentration are essential for microphysical cloud research. Holographic imaging is ideal for three-dimensional analyses of particle size, shape, and spatial distribution in large sample volumes, but its post-processing often leads to operator-dependent results and introduces uncertainties in detection efficiency. Here we present *CloudTarget*, a set of chrome photomasks with a customized pattern of opaque circles, serving as a verification tool to quantify detection efficiency and evaluate size and position errors. CloudTarget provides a ground truth for optimizing hologram processing parameters, including detection, sizing, and classification thresholds, and it facilitates evaluations of size- and position-dependent detection efficiency and uncertainties. Additionally, we present a Convolutional Neural Network (CNN) for object classification that achieves high accuracy with moderate training data. In a holography setup featuring a  $5120 \times 5120$  pixel imaging sensor, a 3  $\mu$ m effective pixel size, and 355 nm illumination, the CNN achieves over 90% recall and precision for particles larger than 7  $\mu$ m in a  $10 \times 1.3 \times 1.3$  cm<sup>3</sup> detection volume. The average focus position error remains below 150  $\mu$ m (1.5 times the reconstruction resolution) for particles 

<sup>&</sup>lt;sup>2</sup>Faculty of Physics, University of Göttingen, Friedrich-Hund-Platz 1, 37077 Göttingen, Germany

<sup>&</sup>lt;sup>3</sup>Institute of Computer Science and Campus Institute Data Science, University of Göttingen, Germany

<sup>&</sup>lt;sup>4</sup>Laboratory of Atomic and Solid State Physics, Cornell University, 523 Clark Hall, Ithaca, NY 14853, USA

post-processing of holograms the data output (particle count, positions and size) underlies accuracy limitations that need to be assessed. Here, we focus specifically on the use of in-line holography for in-situ measurements of warm clouds, where spherical cloud droplets are measured. The basic principle of holography is to illuminate a sample volume with collimated coherent light (i.e. a laser light with a sufficiently large coherence length) and to then record the diffraction patterns of all the objects within the sample volume, i.e. the volume between the last optic of the illumination setup and the image plane of the camera. From that diffraction pattern, the position, and cross-sectional shape of the particles in the sample volume can be determined through complex post-processing of the holograms. Typically, the wavefront is reconstructed in a number of planes parallel to the camera sensor along the optical axis (z-axis in our convention) covering the whole sample volume. In case of sampling warm clouds, the cloud droplets can then be identified as dark sharp circular objects in the amplitude signal in their focus plane. While particles in focus can be found in the planes, each plane still contains all the diffraction patterns from other objects as noise. The reconstructed planes are processed with a threshold to find dark objects. Due to the high noise, typically, a classification is needed to identify droplets and classify the dark objects in the planes into "particle" or "artifact". Other sources of noise such as laser imperfections, camera noise, optical artifacts and, especially for in-flight holograms, dust or dirt on the optics and windows make classification even more challenging.

35

With the advancement of camera technology used in holographic instruments, both the sensor size and the frame rate increase (see e.g. the evolution from HOLODEC (Fugal et al., 2004) and HALOHolo(Schlenczek, 2018) to HOLIMO (Henneberger et al., 2013; Ramelli et al., 2020) and Max-Planck-Cloudkite+ (MPCK+) (Stevens et al., 2021)) which significantly increases the amount of data recorded per flight. For example, in the EUREC4A campaign (Stevens et al., 2021) several hundred thousand holograms with 25MB (5120x5120 pixel, 8 bit) each were recorded by the MPCK+ holographic system. While reconstruction and object extraction is fully automated with the HOLOSUITE software package (Fugal et al., 2009), classification remains a challenge. It can be done by manual labeling, with simple rules (circularity, edge sharpness...) or with machine learning algorithms trained on manually labeled data. The machine learning algorithms range from decision trees over support vector machines (SVM) to Artificial Neural Networks. In particular, simpler methods such as decision trees (Schledewitz, 2016; Schlenczek, 2018) and rules (Glienke et al., 2023) work well for a subset of data, but struggle to generalise across several measurement instances (e.g. flight segments). Neural Networks have been successfully implemented for identifying both water droplets and ice crystals in holographic reconstructions (Touloupas et al., 2020). Classification of small water droplets, however, still remains a challenge for these classifiers (Lauber, 2020). For all methods, the manually annotated training and testing on data is the bottleneck. In light of the increasing amount of data generated by the holographic imagers, an improved and less expensive classification method must be developed that requires a moderate amount of manually labeled training data.

Hong et al. (2024) give a comprehensive overview over traditional, inverse and machine learning approaches for the full processing of holograms containing particles. The mean particle size can directly be extracted through hologram self-correlation (Denis et al., 2006). Additionally, there is ongoing research how to skip the computationally expensive reconstruction algorithms altogether and implement wavefront reconstruction with machine learning or extract individual particle locations and shapes directly from the diffraction patterns in the holograms with Neural Networks (Wang et al., 2018; Ren et al., 2019; Zhang et al., 2022; Wu et al., 2021; Shao et al., 2020; Chen et al., 2021; Schreck et al., 2023; Paliwal et al., 2025). These ground-

breaking approaches have the potential to significantly reduce computational time. However, to make progress now here we focus on the classic approach of wavefront reconstruction with subsequent classification.

Besides generalisation, another issue with current classification methods is that their evaluation is often limited to comparison with manual labels. The inherent inaccuracy of this labeling remains an unknown factor. Furthermore, this evaluation can only verify the classification accuracy, i.e. the extent to which objects found by the reconstruction and object extraction have been misclassified by the classification algorithm. The overall performance of a holographic system and the post-processing of the hologram are still mostly unexplored. In a recent paper by Wu et al. (2024), a printed target with two-dimensional circular objects was introduced for verification of z-position accuracy and as an indication for detection. However, the analysis of the printed target measurements was limited, exact detection parameters like recall were not calculated, although such a target is optimal for analysing them as a function of size and position.

The challenges in analyzing in-situ holographic measurements of cloud droplets include developing a classification algorithm that is fast and accurate, as well as validating the quality of the resulting data against rigorous tests. Key questions that arise during this process include: How accurate are the classifier's predictions? What proportion of the actual particles are detected? Are there false positives — particles identified that do not actually exist? Can the measured size and position of the droplets be trusted? How do the system's performance metrics vary with changes in particle size, position, and concentration?

While users of holographic systems are aware of potential instrumental and processing inaccuracies (e.g., (Glienke et al., 2020)), existing evaluation methods (summarised in section 3.1) are insufficient to fully and quantitatively verify the accuracy of hologram processing. To address this limitation, we developed a verification tool, CloudTarget, designed to enhance the evaluation capabilities for hologram processing and classification accuracy. The CloudTarget uses one or more chrome photomasks with a customised pattern of opaque circles with diameters similar to those of drops in warm clouds. In addition, we present a Convolutional Neural Network (CNN) approach for classifying holograms of particles with circular cross-sections and a minimum diameter of twice the effective pixel size. We evaluate the robustness of the CNN and quantify its accuracy using both laboratory and in-situ holograms collected during EUREC<sup>4</sup>A field campaign Stevens et al. (2021) captured with the Max-Planck-CloudKite<sup>+</sup> (MPCK<sup>+</sup>) holography system. Our results highlight the critical role of selecting appropriate training data for the CNN and demonstrate the unique advantages of CloudTarget for assessing the entire hologram processing workflow, including classification. By utilizing CloudTarget as a controlled test target within the sample volume, we analyze detection performance in terms of precision and recall across all variables such as cross-section position (*x-y*), depth position (*z*), and particle diameter. Additionally, we explore the potential for determining sizing and positional accuracy to a certain degree using CloudTarget.

# 2 Hologram Processing

Figure 1 provides a comprehensive overview of our hologram processing steps. Our hologram processing follows the methods presented by Fugal et al. (2009) and the HOLOSUITE software package (Fugal et al., 2009; Shaw et al., 2012; Schlenczek, 2018).

Figure 1. Schematic showing the post-processing steps of holographic images to extract particle position and sizes from a hologram. The hologram is **acquired** by illuminating the sample volume and recording the diffraction pattern. As an example, we show a real in-situ hologram recorded with the MPCK<sup>+</sup> in a precipitating cloud during the EUREC4A campaign. The static background of the holograms is removed and additional noise **filtering** is applied. Then the hologram is propagated along the z-axis for **reconstruction**. Dark regions are identified using a global threshold, which are then aggregated into "objects". The z-plane focus of the objects are determined and a **classifier** sorts the objects into the classes "particle" or "artifact". In the final step, the particles are **sized** by converting the focused image crop of the particles into binary form and computing the area equivalent diameter. The methods are based on the software package HOLOSUITE (Fugal et al., 2009) with custom extensions and modifications by us.

# 90 2.1 Acquisition

In Figure 1 the **Acquisition** step illustrates the principle of in-line holography. An expanded, collimated laser beam (or an alternate coherent light source) illuminates a sample volume. Within the context of cloud physics, cloud droplets fill the sample volume (terms "droplets" and "particles" used interchangeably throughout this paper). The light is scattered at these droplets, resulting in a diffraction pattern visible in the image plane. In our convention, the z-axis is along the laser beam and the image plane is at z = 0 cm. The directions parallel to the image plane are denoted with x and y. Mathematically, the diffraction in our holograms can be described with Fraunhofer diffraction. The Fraunhofer condition is met for most of the droplet sizes (order of  $10 \mu m$ ) for z larger than a few cm. As shown in Tyler and Thompson (1976) the diffraction pattern for an object with circular cross-section and radius of a at distance z illuminated with coherent light of wavelength  $\lambda$  can be described with

$$I(r) = 1 - \frac{2\pi a^2}{\lambda z} \sin\left(\frac{\pi r^2}{\lambda z}\right) \frac{2J_1(m)}{m} + \frac{\pi^2 a^4}{\lambda^2 z^2} \left(\frac{2J_1(m)}{m}\right)^2 \tag{1}$$

where  $m = \frac{2\pi ar}{\lambda z}$  and  $J_1$  is the Bessel function of the first kind of order 1. I(r) describes the intensity in a radial distance r from the objects center. From the equation, we can infer the difference in diffraction pattern for large vs. small and near vs. far droplets as indicated in the schematic in Figure 1 Acquisition. Droplets of larger diameter (a in Eq. 1) have a higher amplitude in the radial intensity signal. Consequently, even in high noise conditions, their stronger signal can be detected. For droplets closer to the image plane (small z), the high amplitude signal is more localised at small r i.e. the information does not spread to large x-y-extent. This phenomenon can affect detection for particles at large z and at x-y-position far from the center as only a fraction of the signal of the diffraction pattern is recorded by the sensor. In scenarios where a hologram is sparsely populated with small objects in the far-field, such as in-situ cloud droplet holography, the hologram can be approximated as a superposition of the individual droplet diffraction patterns (Eg. 1). The image plane is either directly a camera sensor or the image plane of a magnification lens translating the image onto the camera sensor. As an example, we show a hologram from the 110 MPCK<sup>+</sup> holography system recorded inside a warm cloud during the EUREC4A campaign (Stevens et al., 2021). This image contains strong background signal primarily due to dust and water droplets on the laser window (at high z, see Fig. 2 F) and the camera window (at low z). Additional systematic influences on the recorded hologram are caused by, for example, laser beam inhomogeneities, contaminated imperfect optics within the instrument, possible mechanical vibrations of the optical elements, shot noise from the camera, and varying sensitivity of the camera pixels themselves.

## 2.2 Filtering, Reconstruction and Classification

115

The static background of the hologram is removed (**BG** + **Filtering** in 1). At a detector length / width of 15 mm and an image acquisition rate of 75 Hz of the MPCK<sup>+</sup>, particles in a flow of more than 1.15 m/s, a condition met for all MPCK<sup>+</sup> in-situ measurements, will not appear in more than one hologram. This enables us to eliminate most of the hologram background by calculating the pixel-wise brightness median of neighbouring 2i holograms and perform a pixel-wise division

$$I_{x,y,n}^f = \frac{I_{x,y,n}}{\text{median}(I_{x,y,n-i}...I_{x,y,n+i})},$$
(2)

 $I_{x,y,n}$  is the intensity recorded at pixel x,y in hologram n and  $I^f$  is the intensity after background filtering. We discovered that slight relative movements between the camera and laser can make this background removal non-optimal. Shifting the neighbouring images by a few pixels in x-y to enhance the correlation with the actual hologram improves background removal

$$I_{x,y,n}^{f} = \frac{I_{x,y,n}}{\text{median}\left(I_{x+\Delta x_{n-1},y+\Delta y_{n-1},n-i}...I_{x+\Delta x_{n+1},y+\Delta y_{n+1},n+i}\right)}.$$
(3)

125

130

Typically, the background-removed hologram would then be reconstructed (as explained below). However, through extensive testing we found that the following steps in the processing yield more accurate results when the hologram is further pre-processed and additional noise (e.g. dominant frequency in background removed hologram) is removed. These additional filtering steps were implemented carefully to prevent removal of any droplet diffraction pattern signal. Here, we filter out dominant frequencies in the energy spectrum of the image (visible as diagonal pattern in background removed hologram in 1) and perform Fourier-denoising to filter out Gaussian noise. The optimal noise filtering is, however, likely dependent on holography system used.

Subsequently, the filtered hologram is normalized and then reconstructed by propagation of the wavefront along the z-axis: From the recorded wavefront  $u(z_0)$  at the sensor position  $z_0$  the wavefront  $u(z_1)$  in a x-y-plane at distance  $z_1$  can therefore be calculated, with the angular spectrum method (see e.g. chapter 3.10 in Goodman, 2005). The **Reconstruction** is via angular spectrum method, i.e. implemented in Fourier space, is performed with the a Huygen-Fresnel kernel in filtering form, as explained in Fugal et al. (2009) Eq. 2-5 without additional frequency filters. The wavefront reconstruction provides the amplitude and phase at z-distances throughout the sample volume. For MPCK+ holograms, the reconstruction is performed in  $\Delta z = 100 \, \mu \text{m}$  steps. The  $\Delta z = 100 \, \mu \text{m}$  was empirically found to be optimal for wavelengths of 355-532 nm and a pixel pitch of few microns. For larger  $\Delta z$  the particle may not appear in focus in any of the z-planes, whereas smaller  $\Delta z$  does not lead to higher z-position accuracy while increasing computational effort.

Next, a global threshold, g-threshold in Fig. 1, for each plane is applied to the amplitude reconstructions to extract objects. For calculating the detection g-threshold, a parabola is fitted to the brightness histogram of the actual reconstructed slice (details in Schlenczek (2018) chapter 4.1.2). Patches with n pixels darker than this threshold in m z-planes are saved. Typically n=2, which puts a lower limit of diameter of detectable particles at about 6  $\mu$ m. We refer to these 3D volume crops around dark areas, containing amplitude and phase, as "objects". The focal plane of each of these 3D objects is determined by finding the z-plane with maximum standard deviation of the Sobel amplitude gradient and maximum of the standard deviation of the complex image gradient (Fugal et al., 2009; Schlenczek, 2018).

Considering the whole volume is typically filled with particles/droplets, each reconstructed plane contains not only the particles that are in focus but also signals from the diffraction patterns of non-focused particles and noise or background signal that was not properly removed. Therefore, not all the dark objects can be presumed to be particles. The objects need to be classified into "particles" and "artifacts". As mentioned in prior work, this classification was conducted through manual labeling, simple rules (Glienke et al., 2023), decision trees (e.g. *carft* (Schledewitz, 2016; Schlenczek, 2018)), support vector machines (Ramelli et al., 2020), Neural Networks (Touloupas et al., 2020; Lauber, 2020) or a combination (Glienke et al., 2023). We developed a

Convolutional Neural Network for **Classification** that sorts each object into the two classes based on the amplitude and phase image in focus. Full details of the CNN are provided in section 2.3.

In the past, we have also tried to optimise the focus finding algorithm with the help of machine learning. For training data we used manual label of focus plane. This manual labeling is, however, even more time-costly than the class annotations and therefore less training data was available. The machine learning approaches were not able to achieve higher accuracy on average than the gradient approach (the error in z-focus improved for some objects and got worse for others). After improving the background and noise filtering, we find high accuracy for focus/ z-position detection (see section 5.4).

As a final step, an object-specific threshold (s-threshold in Fig. 1 **Sizing**), is applied to every amplitude image of objects classified as particle. The threshold is calculated as described in Schlenczek (2018) chapter 4.1.3. The particle image below the threshold is dilated, eroded and holes are filled. From this binary image, we determine the size of the particle with the area equivalent diameter, *x-y*-position from the centroid. After all of these processing steps the 3D position and size of all particles in the holography sample volume are extracted.

# 2.3 Improved Classification with a Convolutional Neural Network




We employed a Convolutional Neural Network (CNN) for classification of holography objects (see Figure 1 Classification). CNNs are a class of machine learning algorithm. They consist of various layers inspired by the neuronal structure in brains that process and then feed information into the successive layer. CNNs excel at object detection in images, can learn distinctive image features such as edges and textures and can classify images into different classes (Krizhevsky et al., 2012). The general concept here is to use a Convolutional Neural Network to classify the objects revealed in the reconstruction process (see **objects** in Fig. 1). The aim is to achieve a classification into the two classes *particle* and *artifact* and therefore the output of the CNN should be a likelihood of the object being in each class. The cutoff up to which an object is considered a particle (Particle Classification Threshold: PCT) can then be decided on. As input, the cropped reconstructed planes around the dark regions – what we call objects – were used. We selected a 30x30x2 layer per object, which corresponds to two images with 30x30 pixels. One image is the amplitude and the other the phase, both from the focus planes of the objects determined in the reconstruction and object extraction of the normalized hologram. Both amplitude and phase image are initially padded with the background value of the crop (90th percentile) to form a square image. These square images are then resized to 30x30 pixels through interpolation. For a majority of objects this means upsizing, downsizing is less common.

For training the CNNs we used the trainNetwork function in MATLAB. The function requires defining the layers of the CNN and the training options. The layers we used can be found in appendix A and the training options in B. The CNN consists of five main layers- an image input layer, a 2D convolutional layer, a batch normalization layer, a Rectified Linear Unit (ReLU) activation layer and a fully connected layer. The convolutional layer uses 40 filters of size 10x10. This architecture is tailored for 2-class classification tasks. All CNNs we compare in this paper, have these same input format, layers and training options. We have evaluated more network configurations than the one presented here, including 3D images where all z-planes (instead of just the focus plane) of an object as input to the CNN. Our findings indicate that the effects of network layers, training

options and modification of input (e.g. learning rate, number of epochs, resizing, 3D input with out-of-focus images, padding) are negligible compared to the effect of choice of training data and, more importantly, background and noise filtering in the holograms. Based on our experience, fine-tuning the characteristics of the CNN classifier can not compensate for a hologram that contains too much noise.

The different datasets used for CNN training are explained in section 4.1. All of them are a set of objects with manual annotations from one or two persons. In total, we compare seven CNNs in sections 5.1.1 and 5.1.2, each trained on a different combination of the datasets described in Table 2.

## 3 Verification of Holographic Droplet Data

A critical part of the hologram processing chain is the verification of the various processing steps. In this section, we first describe the typical verification methods for measuring particles/droplets with holography that are available to our knowledge to illustrate that they are not sufficient for a full verification. Generally, these methods can be categorized into two types. On one hand, there are verification tests that can directly be applied to the in-situ holograms that are supposed to be analysed. On the other hand, it can be useful to record controlled test holograms as they provide more insight about the recorded objects. Afterwards, we describe the newly developed characterisation CloudTarget and its role in addressing and filling gaps in the verification process.

## 205 3.1 Established Verification Methods

#### 3.1.1 Direct Verification with In-situ Holograms

# **Manual Object Classification**




It is a common practice to evaluate the automated classification by comparison to a set of manually annotated objects. For this, a number of objects, ideally representative of the entire range of holograms to be analyzed, are annotated manually by an experienced operator with labels "Particle" or "Artifact" (or more classes). The manually annotated class can then be compared with the predicted class by the automated classification method. This method is strongly dependent on the experience of the operator and the limits of human visual perception. In addition, the operator has to perform a large number of repetitive tasks, which makes the work strenuous and very time-consuming. Depending on desired accuracy of labels an experienced operator can label a few hundred objects in an hour. The main limitation is that only the classification can be verified with this method, the actual detection efficiency (or recall) remains unknown.

# **Inverse Methods**

To verify the sizing of particles or droplets in in-situ holograms it is also advised to compare the chosen sizing method with the sign-matched filter approach described by Lu et al. (Lu et al., 2012). In principle, particle positions and locations are guessed (e.g. an informed guess from reconstruction method) and the expected diffraction signal is compared with the actual signal.

Predicted particle size and position is tuned until they match. This method does not depend on the classic reconstruction of holograms followed by thresholding and pixel counting algorithms and instead tries to match the observed diffraction pattern to its prediction. It therefore gives an independent test of the sizing. To some degree it can also indicate misclassification as the algorithm not converging can indicate a false positive (FP) prediction. The method is very computationally expensive and can therefore only be applied to a subset of predicted particles as a test.

#### **Super-hologram**





Typically, a sequence of holograms contains thousands of images. In most applications, like cloud droplet imaging, it can be assumed that on average the droplets are evenly distributed within the sample volume and the size distribution should also be position independent. Therefore, a common technique to find biases in detection efficiency is calculating a super-hologram, where all particles (i.e. after classification and exclusion of all artifacts) from a large number of holograms are placed inside a single sample volume (e.g. (Beals, 2013; Larsen and Shaw, 2018)). With this method, one can determine the *relative* detection efficiency of droplets of different sizes within the volume as a function of 3D-position. Based on this, an effective sample volume and particle size range can be chosen, such that detection efficiency is approximately constant. However, the absolute value of that detection efficiency remains unknown, i.e. it can only provide a relative indication of the detection efficiency within the sample volume.

## **Instrument Inter-comparison**

If other instruments have measured the same particles as the holographic instrument in the in-situ experiment, an intercomparison is possible. Typical for simultaneous measurement of cloud droplets are Cloud Droplet Probes (CDPs) (Glienke et al., 2023; Schlenczek, 2018; O'Shea et al., 2016). The average droplet concentration and size distribution can be compared. Therefore, a low detection efficiency, a high number of falsely classified particles or a clear bias in sizing could be identified if large deviations are found.

The unique feature of holographic instruments are, however, the large sample volume and rate. This is unmatched by conventional droplet probes and therefore a comparison can only be made if averaged over a sufficiently long time, which translates to long flight distance. But especially for cloud measurements, it can not be assumed that concentration and size distribution are constant as shown in Allwayin et al. (2024).

#### 3.1.2 Processing Verification with Test Holograms

## **Resolution Test Target**

To test the resolution of a optical system, including a holographic instrument, a simple method is using a optical test targets such as the 1951 USAF Resolution Test Chart (used in e.g. (Spuler and Fugal, 2011; Beck et al., 2017; Ramelli et al., 2020)). This chart consists of opaque lines of a certain thickness d spaced by d. The thinnest and closest lines with  $d_{min(res)}$  that are distinguishable determine the systems resolution limit. While this test is a first step to check the resolution of the optical system, it does have limitations. The lines with  $d_{min(res)}$  that indicate resolution limit of the target can be placed at only one

position in the whole sample volume for each hologram. Therefore, the position dependence in all 3 dimensions can only be captured by a substantial number of test holograms. The resolution of two thin lines close to each other also does not directly translate to round objects, such as cloud droplets typically measured with holographic systems. Overall, the USAF Chart is a decent tool to check whether the system approximately follows the theoretically expected resolution.

# 260 Test Beads/ Generated Droplets






Another useful test for holographic systems involve glass beads with specified diameters that are brought into the sample volume (Pu et al., 2005). The advantage of this method is that glass beads are optically similar to cloud droplets. In addition, if beads are dispersed throughout the entire sample volume and the concentration is comparable to those found in clouds, the resulting hologram is almost identical to one recorded from cloud droplets, only excluding the in-situ specific noises. However, the characterisation remains limited since the location and total number of glass beads are not known, hence precise validation of individual droplet sizes is not possible. With mono-disperse beads the sizing method can be evaluated by determining the uncertainty in sizing. But since typically concentration is unclear, the glass bead test can not determine the detection efficiency for different sizes/ positions. The bead/ droplet size resolution limit can be obtained by probing different mono-dispersed glass beads. However, in our experience, the smaller the glass beads, the more they tend to clump together. As at least parts of the clusters are classified as particles in our processing, this makes it increasingly challenging to identify the measured sizes corresponding to individual beads instead of bead clusters. Consequently, this method is not ideal for sizing beads with diameters below 15 µm, which is a critical size range for the majority of cloud droplets. The use of ultrasonic dispersion techniques, such as those described in Giri and Berg (2023), may improve the separation of beads and enhance the method's applicability in this regime. When using poly-disperse glass beads with a known size distribution comparing the given size distribution of the glass beads with the measured size distribution, the relative detection efficiency of different sized beads/particles can be estimated. Guildenbecher et al. (2013) improved this method by probing polystyrene beads that are dispersed in oil in a thin cuvette, thereby z-position and concentration are well-defined but similar to our CloudTarget the droplets are only located in a discrete z-plane. Similar principles and arguments apply for using a droplet generator (Ramelli et al., 2020) or droplet injector (Fugal, 2007) to benchmark the sizing accuracy.

# **Modeling Approach**

Another possibility to verify particle detectability is based on generating synthetic particle holograms with optical properties as close to the actual instrument as possible (e.g. (Fugal and Shaw, 2009)). Via this method it is possible to find the volume of uniform detectability based on particle diameter, which can be compared with the results from the super-hologram. A detailed analysis based on applying an instrument model is presented in Schlenczek (2018) chapter 5.1.2. Fundamental limitations of the modeling approach are effects and processes which occur in the actual holographic setup but are not captured by the model (e.g. variation of the hologram background in time, optical aberration,...).

## 3.2 Quantitative Experimental Verification with CloudTarget

As previously mentioned, there are limited options to obtain an in-situ calibration of both particle position and particle size of an in-line holography setup with a realistic number concentration and size distribution compared to atmospheric clouds. We developed the idea of a calibration setup for in-line holography to record holograms of 3D volumes wherein all the *particles* present are well defined, meaning their position and size are known. We call this calibration setup CloudTarget.

## 3.2.1 CloudTarget Design





The calibration setup consists of a metal box designed to hold up to 5 individual glass chrome photomasks (see Fig. 2 D). The photomasks (or "glass targets") are made of Quartz-Fused Silica and have an anti-reflective coating achieving a transmission of 92% for the relevant wavelengths. Each photomask measures  $45 \times 45$  mm (Fig. 2 E). The size was chosen so that it fits into both of the MPIDS' holographic systems with room for shifting it in x-y.

On each photomask around 6000 opaque (chrome coating OD5) circular disks ranging from diameters of 4 to 70  $\mu$ m are printed as shown to scale in Fig. 2 A. The diffraction pattern of a cloud droplet can be approximated with that of an opaque disk as shown in Tyler et al. (Tyler and Thompson, 1976). Consequently, a hologram recorded with CloudTarget in the sample volume approximately looks like a hologram recorded in clouds, with the exception of the particles being located at discrete z-positions only. The CloudTarget presents a greater challenge compared to real clouds due to significant interference from densely packed objects within the setup. However, as a controlled laboratory environment, it offers a cleaner background than what is typically encountered in real clouds. The size distribution of the particles printed on the photomasks of CloudTarget, is shown in Fig. 2C in orange compared to the size distribution in the CLOUD-test dataset in blue which is a randomized subset of particles from in-situ cloud holograms. The size distribution mimics what can typically be found in clouds with an over-representation of large particles (>20  $\mu$ m) to ensure sufficient statistics for calculating size-dependent verification metrics. The printed particles are distributed with a Poisson disk distribution with a minimum distance of 390  $\mu$ m to avoid overlapping and excessive clustering in the x-y plane.

Each CloudTarget contains a unique identifiable pattern in the centre (highlighted in orange in Fig. 2B). This pattern can be used to determine the relative *x-y* displacement of the target with respect to the camera sensor, the rotation and whether the target is flipped or not. The design ensures that this pattern is captured on every hologram, even if the target is not exactly centered in the holographic sample volume. With that, the objects classified as particles can be matched one-to-one with the CloudTarget dataset. Thus, CloudTarget can be used effectively as *ground truth* for particle size, concentration and *x-y-z* position with respect to the camera sensor.

The brightness of the laser in the holographic system must be adjusted depending on the number of photomasks used in CloudTarget, due to the limited transmission caused by reflection at the interface of the individual CloudTarget photomasks. We have found that the number or order of targets has no effect on CNN precision or recall if the number of targets is below three. If more than three photomasks are used, the signal-to-noise ratio deteriorates due to low transmission which leads to a lower mean brightness and reflections become visible. Therefore, in the tests described in this paper only a single photomask

**Figure 2.** Overview of CloudTarget. **A** show the distribution of the 6000 printed particles on a photomask of CloudTarget; **B** zoom into center of photomask with orange circles indicating the tracking pattern, a specific, pre-defined pattern used to align measured data with the ground truth; **C** histograms of the particle sizes in the printed particles and the size distribution of the random in-situ dataset CLOUD-test;**D** the target holder can hold up to 5 photomasks to be measured at once inside the sample volume; **E** photo of one of the photomasks adapted from (Stieger, 2024); and **F** schematic of how a test hologram is recorded with CloudTarget.

was used in CloudTarget, which was altered in z-position. The z-positions of the photomask is given as the mean reconstructed distance of the identified particles from the image plane throughout section 5. The change in optical path length due to the refractive index of n = 1.46 of the 2.3 mm thick photomask is approximately 1 mm and neglected here.

# 3.2.2 Experimental Procedure

In the following, we describe the procedure used to record and analyse test holograms with CloudTarget. The fundamentals of CloudTarget are described in Stieger (2024). First, the photomasks are placed within the target holder. The CloudTarget is then brought into the sample volume of the holographic setup under test as depicted in Fig. 2F. During hologram recording, the target is moved continuously in *x-y* plane while ensuring that the edges of CloudTarget remain outside of the sample volume. This results in the printed particles being at different *x-y*-positions in each recorded hologram, thereby allowing the use of typical background removal methods like division of median of neighbouring holograms in a sequence.

After acquisition, the holograms are processed with the processing chain explained earlier, from background removal to classification and sizing, i.e. exactly the same procedure is used as for the in-situ holograms. In the following analysis step, Cloud-Target is considered ground truth. We know the particle sizes and positions and therefore concentration and size distribution (with imperfections as explained below) a priori. This ground truth is compared with the measured objects that are classified as particles.

As mentioned, each CloudTarget contains a specific particle pattern in the center that allows us to find the x-y-shift, rotation and orientation of CloudTarget with respect to the image plane. Even after translation and rotation of the ground truth data to overlay with the experimental data, we see misalignment towards the edges of the hologram. This misalignment seems to be roughly consistent for each photomask over different holograms but varies for the different photomasks used. We therefore suspect that these differences are due to the manufacturing tolerances used to produce the photomasks of CloudTarget, which makes it difficult to use CloudTarget as an exact ground truth for the x-y-position (more on this in section 5.4). To improve particle matching for sizing and detection analysis we further correct the ground truth position by using a two-dimensional Particle Image Velocimetry (PIV) algorithm (as described in Stieger (2024)). This adjustment leads to a better alignment of the ground truth with the measured particles. With the CloudTarget holograms, for which one photomask was used, we expect the z-position of all the found particles to be constant. However, in every CloudTarget hologram, we find the main z-plane with all particles on (sometimes slightly tilted in x-z and/or x-y) but also a parallel plane in about 1.5 mm distance in z where particles have been predicted. We found that this layer contains ghost particles, which we believe are caused by reflections within the photomask. Since this is only an artifact of the test holograms with the photomasks and would not occur in real in-situ holograms, this layer of ghost particles is manually removed before further analysis (see (Stieger, 2024) for a more detailed analysis and justifications).

To evaluate the detection performance of the holographic system and subsequent processing, we count the predicted positives PP in the measured data (all objects with classification value larger than the chosen Particle Classification Threshold) and the positives P (particles) in the ground truth data. If a measured particle is close enough to a ground truth particle (less than 3 pixels deviation in x and y) and has approximately the same size (less than 2 pixels deviation), mathematically expressed as  $|x_m - x_{gt}| < 10 \, \mu \text{m} \, \wedge \, |y_m - y_{gt}| < 10 \, \mu \text{m} \, \wedge \, |d_m - d_{gt}| < 6 \, \mu \text{m}$ , it is counted as a true positive TP. The number of TP is robust against varying these limits. All the PP that are not TP are defined as false positives FP and all the P that are not TP are false negatives FN.

#### 3.3 Summary and Comparison

In Table 1 we give an overview about the verification methods described in section 3.1 and CloudTarget described in section 3.2. Check marks indicate the verification capabilities. The resolution limit, i.e. the smallest resolvable diameter can be measured with the USAF target (with the constraint that resolution of lines might be clearer than resolution of circular objects) and CloudTarget directly. Through instrument inter-comparison, it can also be estimated from comparison of measured size distribution. Classification methods can be compared and evaluated against manual labels, the inverse method can confirm it for a small set of objects and with the ground truth from CloudTarget test holograms it can fully be tested. For testing the sizing algorithm, one can measure mono-disperse particles with a known size in a test hologram like glass beads or droplets from a droplet generator, use the inverse method to measure size of selected particles threshold-independently, or compare the measured sizes to the CloudTarget ground truth (although we will show that there is a bias in sizing CloudTarget objects in test holograms compared to sizing droplets in in-situ holograms, see section 5.3). While, to our knowledge, a verification for

|                           | in-situ holos |              |          |             | test holos |              |              |
|---------------------------|---------------|--------------|----------|-------------|------------|--------------|--------------|
|                           | m. labels     | inv.         | S-holo   | inst. comp. | USAF       | glass beads  | CloudTarget  |
| Resolution limit          |               |              |          | (√)         | ✓          |              | ✓            |
| Classification            | ✓             | <b>(√)</b>   |          |             |            |              | $\checkmark$ |
| sizing                    |               | $\checkmark$ |          | <b>(√)</b>  |            | $\checkmark$ | (✓)          |
| detection                 |               |              | relative | relative    |            |              | $\checkmark$ |
| focus accuracy            |               | $\checkmark$ |          |             |            |              | (✓)          |
| x- $y$ -position accuracy |               | <b>(√)</b>   |          |             |            |              | <b>(</b> ✓ ) |

Table 1. Overview of verification methods: manual object annotations (m. labels), inverse methods (inv.), super-hologram (S-holo), instrument inter-comparison (inst. comp.), USAF resolution test target (USAF), glass beads and CloudTarget. Each verification method can validate different aspects of the processing chain or different quantities obtained from it. The applicable verifications for each method is marked with a checkmark symbol  $\checkmark$ . Certain methods can not fully verify a measurement or provide absolute values on the accuracy but can indicate towards accurate or inaccurate measurements. These are marked with a checkmark symbol in parentheses ( $\checkmark$ ). The resolution limit and absolute detection are discussed in section 5.2, classification in sections 5.1.1 and 5.1.2, sizing in section 5.3 and focus accuracy and x-y-position accuracy in 5.4.

absolute detection efficiency i.e. determining recall of the whole system, was missing until the introduction of CloudTarget, it was possible to quantify the relative detection efficiency as a function of particle position or size with the super-hologram method and through instrument inter-comparison. The current version of CloudTarget can verify position accuracy only to an extent. We can measure the z-position scatter and therefore a relative mean error. Verifying *x-y*-position revealed potential inaccuracies in the design of CloudTarget itself, but should theoretically be possible (see section 5.4 for more details). Otherwise the inverse method can be used to determine particle position independently of the processing algorithm. It can, however, not reveal position inaccuracies based on misaligned optics (only errors stemming from the processing algorithm) as it assumes perfectly aligned optics.

Overall, from Table 1 it is clear that CloudTarget is a powerful verification tool for holographic systems and fills gaps in the validation other methods were not able to cover yet.

## 380 4 Methodology

#### 4.1 Datasets for Classifier Training and Experimental Verification

The datasets used throughout this paper can be divided into two groups: datasets that were used to train the classification CNN (section 2.3) and datasets to verify the holographic methods. The CLOUD-XX datasets for CNN training each contain a number of objects that were chosen from a given number of holograms  $N_{holos}$  and then manually annotated by one or two operators

| Use          | Dataset    | Туре       | Established Variables                      | $N_{holos}$ | $d_{\mathrm{mean}}, d_{\mathrm{p95}}$                                   | Notes                          |
|--------------|------------|------------|--------------------------------------------|-------------|-------------------------------------------------------------------------|--------------------------------|
| CNN Training |            |            | labala 1 Oparator                          |             | n: 14um 24um                                                            | 2 flights                      |
|              | CLOUD-3k   | in-situ    | labels 1 Operator:<br>P, A, LP, LA, G, OOF | 1500        | p: 14μm, 24μm<br>(o: 15μm, 26μm)                                        | 3k objects                     |
|              |            |            |                                            |             |                                                                         | 22% particles                  |
|              | CLOUD-8k   |            | labels 1 Operator:<br>P, A, LP, LA, G, OOF | 2200        | p: 14μm, 23μm<br>(o: 15μm, 25μm)                                        | 2 flights                      |
|              |            | in-situ    |                                            |             |                                                                         | 8k objects                     |
|              |            |            |                                            |             |                                                                         | 22% particles                  |
|              |            |            |                                            |             |                                                                         |                                |
|              | CLOUD-l    | in-situ    | labels 1 Operator:<br>P, A, LP, LA, G, OOF | 2300        |                                                                         | 2 flights                      |
|              |            |            |                                            |             | $23\mu \text{m}, 36\mu \text{m}$ (o: $25\mu \text{m}, 46\mu \text{m}$ ) | 9k objects                     |
|              |            |            |                                            |             |                                                                         | 4% particles                   |
|              |            |            |                                            |             |                                                                         | >20μm                          |
|              | CLOUD-0    | in-situ    | labels 1 or 2 Operators:                   |             | $19\mu\mathrm{m},30\mu\mathrm{m}$                                       | prelim. filtering              |
|              |            |            |                                            | 126         |                                                                         | 9k objects                     |
|              |            |            | P, A, OOF                                  |             | (o: $16\mu m$ , $30\mu m$ )                                             | 57% particles                  |
|              | CLOUD-5h   | in-situ    | labels 1 Operator:<br>P, A, LP, LA, G, OOF |             | $15\mu \text{m}, 23\mu \text{m}$ (o: $15\mu \text{m}, 25\mu \text{m}$ ) | filter tests                   |
|              |            |            |                                            | 5           |                                                                         |                                |
|              |            |            |                                            |             |                                                                         | 122k objects                   |
|              |            |            |                                            |             |                                                                         | 8% particles                   |
|              | CLOUD-test | in-situ    | labels 2 Operators:                        | 500         | p: 15 $\mu$ m, 24 $\mu$ m                                               | 2 flights                      |
|              |            |            |                                            |             |                                                                         | 1.5k objects                   |
|              |            |            | P, A, LP, LA, G, OOF                       |             | (o: $14\mu m$ , $23\mu m$ )                                             | 38% particles                  |
|              |            |            |                                            |             |                                                                         | no overlap with training holos |
|              | CLOUD-inv  | in-situ    | size via (Lu et al., 2012)                 | 60          | $14\mu\mathrm{m}$ , $24\mu\mathrm{m}$                                   | 2 flights                      |
|              |            |            |                                            | 00          |                                                                         | 7k predicted particles         |
| Verification | TARGET-5   |            | ground truth                               |             |                                                                         |                                |
|              | TARGET-8   | Cloud      | ground truth                               |             | 19μm, 55μm<br>4-70μm                                                    |                                |
|              | TARGET-10  | Target     | ground truth                               | 1           |                                                                         | one photomask                  |
|              | TARGET-17  | test       | ground truth                               | 1           |                                                                         | different z-pos                |
|              | TARGET-19  | holograms  | ground truth                               |             |                                                                         |                                |
|              | TARGET-21  |            | ground truth                               |             |                                                                         |                                |
|              | MICRO1     | microscope | inter-particle                             |             |                                                                         | CloudTarget photomask          |
|              | MICRO2     | image      | distances                                  |             |                                                                         | Keyence VK-200                 |
| C 1          |            |            |                                            |             | 1 10 1                                                                  | CO TO T 1.0 1.0 1              |

Table 2. Overview of all datasets used for this paper used for the training of the classification CNNs and for verification purposes. CLOUD datasets include selected objects, manually labeled as either particles or artifacts, from reconstructions and object extraction of in-situ holograms captured with MPCK<sup>+</sup> during the EUREC4A campaign. The holograms for the CLOUD datasets are reconstructed between z=2 cm and z=18 cm. They are used for CNN training and verification against manual labels. For the CLOUD datasets the sizes of only particles "p" and objects (particles and artifacts) "o" are given separately. TARGET datasets are holograms recorded with CloudTarget at different z-positions from the image plane, e.g. TARGET-5 is the dataset where the photomask was at a distance of 5 cm from the camera image plane. The MICRO datasets are microscope scans of the photomask used in CloudTarget.

(as indicated in Table 2). Manually labeling the objects from holographic reconstruction and object extraction, as done for the CLOUD datasets, is a time-consuming effort. For labeling, the operator examines the amplitude and phase image of the object as well as the image gradient along z and can therefore make a decision on "particle" or "artifact". In some datasets, we added the classes "LA: likely Artifact" and "LP: likely Particle" to distinguish cases where the operator is unsure to cases where the operator is convinced about the label. "OOF: out-of-focus Particles" are particles, where the focus finding algorithm did not find the optimal focus and "G: Ghosts" are ghost particles that look like round particles but are optical artifacts.

CLOUD-3k, CLOUD-8k and CLOUD-1 were selected from holograms taken from two MPCK<sup>+</sup> flights in the EUREC4A campaign ((Stevens et al., 2021)). Holograms were randomly selected from the whole domain and processed using our final background processing and filtering methods prior to reconstruction. For CLOUD-3k and CLOUD-8k the objects where chosen completely randomly from the reconstructions and object extractions, their size distribution and fraction of particles/artifacts are representative of the EUREC4A holograms. The CLOUD-1 dataset solely contains objects >20 µm from the same randomly selected holograms. Additionally, two other datasets that were labeled before or during the optimisation of background removal and noise filtering were used. For the CLOUD-0 dataset, only static background removal was performed as the hologram filtering method – no additional noise filtering was applied as this dataset was created before our improvement of the filtering. The dataset consists of randomly selected objects with an addition of large objects all taken from few holograms from short selected sections of one EUREC4A flight. The CLOUD-5h dataset consists of labeled objects and was originally used to determine the optimal background and filtering method. The objects are chosen from selected regions of the same 5 holograms but processed with 9 different background removal and filtering methods each, including methods that were considered less than ideal. Within each selected region of the different reconstructions of the 5 holograms, all objects were chosen, thereby preserving the size distribution and fraction of particles/artifacts.

395

400

For the CLOUD-test data, we randomly selected objects from 500 holograms from two chosen MPCK<sup>+</sup> flights in the EU-REC4A campaign. Since this dataset is used for verification (see section 5.1.1) they were selected from holograms that are not part of any of the training sets. The objects for verification in CLOUD-test were labeled independently by two operators. Each operator was asked to classify all the objects into five groups, namely "likely artifact", "sure artifact", "likely particle" and "sure particle" and "ghost". Objects labeled as ghosts were removed from the test and training sets. Ghosts are optical artifacts that look similar to particles and they are therefore hard to distinguish but identification by the classifier is not needed as ghosts can easily be removed from in-situ data due to their reoccurring position over many holograms. Especially for the verification dataset CLOUD-test, the operators were asked to label the objects carefully and to not rush. This in combination with having two independent operators should ensure the best possible manual labeling. The agreement between the independent labels two operators was > 98% in the sure classes and on average 96% when including the "likely" labels. The two operator labels of the objects used for validation from CLOUD-test were then combined into a binary classification of "particle" or "artifact" (in case of disagreement between operator labels "sure" always overruled "likely" and if needed the more experienced operator overruled the less experienced).

The TARGET-z datasets contain the data of holograms recorded with MPCK<sup>+</sup> of CloudTarget following the procedure described in section 3.2.2. The holograms are processed in the same way as in-situ holograms, including background removal

and noise filtering. The number in the dataset name indicates the distance of the photomask from the image plane in cm (e.g. TARGET-5 for photomask 5 cm from image plane, particles reconstructed at z = 5 cm.

All holographic datasets are recorded with the MPCK<sup>+</sup> holographic system, which has a sample volume of 1.5 cm  $\times$  1.5 cm  $\times$  22 cm and a wavelength of  $\lambda=355$  nm. The effective pixel size, which is a result of sensor pixel size and magnification lens, is 3 µm. For the CLOUD datasets the z-extent of the sample volume is limited to 2-18 cm. For further verification of x-y-position or inter-particle distances in the TARGET holograms, the photomask used in CloudTarget was recorded using the Keyence VK-200 microscope. For each microscopy dataset the microscope scans a number of neighbouring images with a 10x objective of the field of view of 1.35 mm  $\times$  1.012 mm that is resolved onto  $1024 \times 768$  pixels. For MICRO1  $26 \times 3$  images were scanned and for MICRO2  $10 \times 4$ . The images are then assembled into a an overview of the scanned region, which is saved at lower resolution (MICRO1: 23.3 mm  $\times$  4.1 mm saved in 4199  $\times$  743 pixels, MICRO2: 12.6 mm  $\times$  3.8 mm saved in 9078  $\times$  2753 pixels). Afterwards a threshold is applied to detect the particles and the inter-particle distances are measured. With this processing circles >9 µm printed on the photomask are detected.

Table 2 provides a comprehensive overview of all datasets.

#### **4.2** Evaluation metrics

425

440

450

To evaluate detection and classification performance, we use precision and recall as metrics. The recall is defined as

435 
$$\operatorname{Recall} = \frac{TP}{TP + FN} = \frac{TP}{P}$$
, (4)

where TP denotes the true positives, meaning particles correctly found. FN are the false negatives, particles that were not found and P is the sum of both, so all the real particles in the hologram. The recall is always between 0 and 1, and is a measure on what fraction of the particles present in the probing volume were found by the instrument and associated analysis methods. If the classifier correctly identifies all true particles as particles, the recall rate reaches one. Conversely, if no true particles are classified as particles, the recall rate drops to zero. In the results section two different definitions of false negatives FN are used. When evaluating the classification method against manual labels given to the objects, the false negatives only entail objects that were misclassified. In Figure 3 this is denoted with FN1. However, there are also FN that were completely missed by the post-processing and reconstruction with object extraction (FN2). These can only be identified when there is a known ground truth, as it is in the CloudTarget experiments (see section 5.1.2).

445 The precision is defined as

$$Precision = \frac{TP}{TP + FP} = \frac{TP}{PP} , \qquad (5)$$

where FP is the false positives, so any artifacts that is misclassified as a particle. The predicted positives PP represent all the objects that the classifier identifies as particles, i.e. the definitive output that end users rely on for their research and analysis. The precision therefore measures the proportion of objects classified as particles that are actually particles. If all the particles identified by the classifier are true particles, the precision rate reaches one. Conversely, if none of the classified particles are true particles, the precision rate falls to zero. It represents the ability of the classifier to minimise false positives and provides

Figure 3. Schematic illustrating the relationship between Positives P, Predicted Positives PP, False Negatives FN and False Positives FP. There are two types of Fale Negatives FN = FN1 + FN2: FN1 are particles that were simply misclassified by the classifier. FN2 are particles that were not classified at all because they did not appear in the object list and were overlooked in the reconstruction and object extraction step before classification.

information about the accuracy of its positive predictions. Typically, the goal is to have high recall while maintaining an equally high precision. An ideal scenario—rarely achieved in real-world applications—is when both recall and precision equal 1.0. Combining recall and precision into a single metric can be highly useful, as it simplifies comparisons between different algorithms. This can be expressed with the  $F_1$  score, the harmonic mean of precision and recall

455

460

$$F_1 = \frac{2}{\text{Precision}^{-1} + \text{Recall}^{-1}} = \frac{2TP}{2TP + FN + FP}$$
(6)

When analysing the results with the characterisation target it is crucial to note that the size distribution of the particles printed on the targets does not represent common size distributions found in clouds measured by our holography instruments. Specifically, large particles (>20  $\mu$ m) are overrepresented in the targets (as shown in Fig. 2 C). Given that larger particles are easier to detect, we would overestimate precision and recall if we show the average over all particles. To prevent any misinterpretation, we show size-dependent results, wherever possible. Particularly for detection of small droplets we want to resolve the size dependency up to micrometer accuracy. Our sizing error, however, is on the order of at least 1  $\mu$ m for most holograms (see section 5.3). For the sizing accuracy, we calculate the difference of the measured diameter and the ground truth diameter for all TP of the TARGET measurements. Therefore binning our results by the measured size  $d_m$  or the ground truth size  $d_{gt}$  from

CloudTarget are not interchangeable. While we have both a measured and a ground truth size for the TP, the FN were missed by our processing algorithm and therefore no size was measured and the FP are not actual printed particles and therefore we can not attribute a ground truth size with them. To not mix binning of ground truth size and measured size in the calculation of recall Eq. (4) or precision Eq.(5) we use the ground truth diameter  $d_{gt}$  for binning recall and the measured diameter  $d_m$  for binning precision results. This also allows a physical interpretation of the binned precision and recall. Precision is interpreted as how many of the measured particles of a certain measured size are actual particles. Recall is interpreted as how many of the particles with a certain actual size are detected by the holographic system and the post-processing.

This distinction between measured and ground truth size is not needed when testing the CNN classifiers by comparison with the manually labeled dataset. There are no ground truth values for the diameter of the particles in CLOUD-test, simply because the ground truth is not known. In this analysis, FN only covers the particles that were incorrectly classified as artifacts but were still identified by the threshold after reconstruction and are part of the object set. In 3 this is denoted with FN1. As mentioned above, this is a limitation of this verification step. However, it means that all FN have a measured size  $d_m$  and therefore the binning is performed with the measured size for all detection parameters in section 5.1.1.

#### 5 Verification Results


## 480 5.1 Neural Network Classification

# 5.1.1 Neural Network Classification Compared to Manual Annotations

In this section, we evaluate the proposed CNNs, that are trained on different datasets, based on comparison to manual annotations as described in 3.1.1 using Cloud-test dataset (see Table 2). As described in section 4.1 the labels used in CLOUD-test are a combination of the labels of 2 operators. These labels are then used as a baseline in this section, but it has to be noted that they are not a real ground truth. For in-situ data a real ground truth does not exist, we have no certain knowledge about the actual number, size distribution and spatial distribution of the probed particles. This comparison to the manual annotations can only verify the object classification. The inter-operator agreement in labeling the objects is shown with the dark grey crosses in Figure 4, where each cross represents the agreement of one operator to the other in terms of precision and recall. The high inter-operator agreement indicates successful noise filtering of the holograms as most objects are clearly distinguishable as "particle" or "artifact". It also indicates only a small operator bias, which is important since the large training sets are mostly annotated by a single operator.

The predicted classes by each of the seven CNNs of the CLOUD-test objects are compared to the operator labels to verify their classification and compare the effect of different training datasets. Based on the equations given in 4.2 precision and recall are calculated for varying Particle Classification Thresholds and shown in Figure 4. It is important to keep in mind, that recall is based on only the false negatives that were misclassified, not the ones that were never found by reconstruction and object extraction (FN1 in Figure 3). Remarkably, precision and recall of all CNN based predictions both exceed 95% for the

Figure 4. Precision-recall curves for comparison of prediction of CLOUD-test set made by CNNs trained with the different CLOUD datasets (see Table 2) to the manual annotations of the CLOUD-test set. Here, precision and recall are an indicator of how well the classifier has worked, it does not consider particles that might have been entirely missed by reconstruction and object extraction. The round marker shows precision and recall for PCT = 0.3 and the square marker for PCT = 0.1. The light grey star shows the optimal performance at precision = recall = 1, where all particles are correctly classified. The grey crosses show the inter-operator comparison: for each cross the manual annotations of one operator were assumed to be ground truth and the precision and recall of the other operators label were calculated. In the inset we show F1-score (Eq. 6) as a function of diameter for a Particle Classification Threshold of 0.3. The low F1-score for the 20  $\mu$ m bin is likely an outlier due to a low number of samples.

optimal Particle Classification Threshold. Even the CNN trained on the CLOUD-o training data, which consists of objects from holograms with non-optimal noise filtering from only a small section of one MPCK<sup>+</sup> EUREC4A flight, can generalise to the test data from the full two flights with the new noise filtering applied. The small random dataset of 3000 random objects (CLOUD-3k) achieves high precision and recall values but a significant improvement can be observed when increasing the number of objects in the training data to 8000 (CLOUD-8k). The general trend can be observed that precision and recall improve further when even more training data is added. The two best performing CNNs are trained on CLOUD-8k combined with CLOUD-1 and CLOUD-0 with P:97%, R:98% and with added CLOUD-5h P:98%, R:98%. As an inset in Figure 4 we


show the size dependent F1-score of the CNN trained on CLOUD-8k-l-o. The F1-score is independent of size for particles of diameter <20 µm (low F1 of last data point is caused one misclassification in small sample). Larger particles are rare, they are not contained in the randomly selected CLOUD-test data.

This test of manual annotations compared to automatic classification is reliable in finding the optimal automated classification method within the human error. As mentioned the test set is manually labeled and has therefore a small inherent error. Moreover, a test set is always limited. Here we decided on 1.5k objects. Considering each hologram can contain thousand of objects and we intend to analyse  $10^5$  MPCK<sup>+</sup> holograms from EUREC4A (Stevens et al., 2021) this of course can never be a fully representative sample.

The precision we found here can be assumed to accurate within these limits of operator errors and the limited number of objects in the test set. However, it must be noted that the evaluation against a manual test set is not optimal for determination of recall. It can only verify the performance of the classifier itself. Any limitations in detection (recall) due to flaws in the setup, the reconstruction or the thresholding and object detection extraction can not be quantified. The recall has to be interpreted as "of the particles that were found by reconstruction x % was classified correctly".

Within the limitations of a classifier that is trained on manually annotated data without a ground truth, the CNNs trained on CLOUD-8k-l-+ perform optimal and there is little to no room for improvements when relying on manual labels. Beyond the excellent performance of the CNN classifiers (as shown here and 5.1.2), a significant improvement is the reduced processing effort in object classification. Previously, we aimed to extract the particle information from the hundred thousand of holograms from MPCK<sup>+</sup> recorded during EUREC4A using decision trees. However, each decision tree was only capable to generalise over  $10^2$  holograms and therefore requiring manual annotation of training data and verification of the tree predictions for each  $10^2$  hologram section. This resulted in an estimated classification output of  $<10^3$  holograms per day. In contrast, the CNN-based approach enables fully automated classification on a high-performance computing cluster, which increased the classification outputs to  $10^4$  holograms per day. Now, despite the initial training data annotations and verification of the CNN, no further operator interaction is needed.

## 5.1.2 Neural Network Classification Compared to CloudTarget Ground Truth







The CloudTarget can be used to confirm the choice of the object classification method and selection of training data for the classification CNN as shown in Figure 5. We show the results for all six TARGET-z holograms combined. Remarkably, the classifiers detect CloudTarget particles which supports the assumption of the printed circles in CloudTarget as a valid proxy for cloud droplets. The performance in terms of precision and recall when testing with CloudTarget is weaker than when testing against the manually annotated in-situ holograms, i.e. CLOUD-test. A small deviation could be explained by a different mean z-position of the combination of the TARGET datasets used here compared to the CLOUD-test set or by the fact that the CNN classifier was trained on manually labeled data and thereby learned the human labeling biases.

A weaker recall is expected nonetheless. As mentioned, when calculating the recall compared to the manual test set the false negatives FN are only the objects that were misclassified as artifacts FN1 in Figure 3. But when comparing to CloudTarget,

we find all the FN = FN1 + FN2 which are objects that were misclassified as artifacts FN1 but also particles that were not found by the reconstruction and never appeared even in the objects list FN2. Therefore, assuming the classifiers perform the same on test holograms as they do on in-situ holograms, the actual recall here has to be smaller than the recall when comparing to manual labels.

The precision in the manual comparison gave us an estimate how precise the classification of the CNN is compared to human labeling. With CloudTarget we see overall worse precision which could indicate that the human labeling itself has a high inaccuracy and artifacts are too often mislabeled as particles by the operators. However, we argue that more ghost particles are present in the TARGET holograms caused by reflection (see section 3.2.2 about ghost layer) that are not identified as artifacts by the classifier but since they would not exist in in-situ holograms we can ignore this fault of the classifier. In reality, the low precision is likely due to a combination of these reasons: human labeling might be slightly biased towards labeling too many objects as particles but the precision of only 80% can be seen as a lower limit as some of the FP are likely caused by the reflections photomask specifically as explained in section 3.2 and the actual precision is likely closer to the one found in 4. Despite that, in Figure 5 we see similar trends in how CNNs performs as those shown in Figure 4 for particles <20 µm, e.g. CLOUD-o and CLOUD-3k perform worse than the others. This confirms the similarity between the in-situ holograms and the test holograms recorded with CloudTarget, supporting the assumption that results from CloudTarget tests are transferable to in-situ cloud holograms.

For objects larger than 20 µm, it is clear that a CNN is needed that is also trained on a substantial amount of large objects (see Fig. 5 right where CNN trained on CLOUD-3k and CLOUD-8k perform worst). CNNs trained exclusively on randomly selected objects is associated with a low recall for objects >20 µm, indicating that it struggles to generalize to larger objects. This occurs despite the fact that the input images for classification are resized to a consistent size (see section 2.3).

# 5.1.3 Choice of Classifier and Particle Classification Threshold (PCT)







We choose the network trained on CLOUD-8k-l-o as the final classification method based on the results shown in 5.1.2 combined with the results from the comparison to the manually annotated test set in section 5.1.1. All results in the following are based on classification with a CNN trained on this dataset.

The Particle Classification Threshold is typically chosen based on optimising F1-score, but can also be chosen differently if either precision or recall are prioritised. In Figures 4 and 5 we indicate precision and recall for two different Particle Classification Thresholds (0.3 and 0.1) with the round and square marker respectively for the chosen classifier. The comparison to manual labels would suggest an optimal PCT of 0.1 (square), whereas recall would drop by 3% (to 95%) and precision only gaining 2% (to 99.7%) when using a PCT of 0.3. When testing with CloudTarget a PCT of 0.3 seems to be optimal, although an even higher one would be best for the largest droplets (>20 µm).

We choose a PCT of 0.3 as a reasonable cutoff as it shows close to optimal performance in both the test against manual labels and in the CloudTarget test. If, as explained in section 5.1.2, the low precision in CloudTarget results is associated with ghost

**Figure 5.** Precision and recall of prediction of CloudTarget holograms with CNNs trained by the different CLOUD datasets compared to the ground truth of printed particles. Left: precision and recall for particles  $<20 \,\mu m$  in diameter so that size distribution is comparable to CLOUD test comparison in Figure 4 (see 2 C). Right: precision and recall for large objects. A CNN that was trained on substantial number of large objects is needed to correctly classify large objects. As in Figure 4 round markers indicate a PCT = 0.3 and square markers for PCT = 0.1.

particles that would not exist in in-situ results, it is justifiable to use an even lower PCT between 0.1 and 0.3 for analysis of in-situ holograms.

## 5.2 Droplet Detection



Detection efficiency (recall) highly depends on size of the droplet and the position within the 3D measurement volume as mentioned in section 2. Further away from the camera (large z-coordinate) and near the edges of the x-y-cross-section it is harder to detect particles, especially small ones.

These trends can be deduced based on the radial frequencies and amplitudes of the diffraction patterns (see section 2 for detailed discussion). The relative detection bias of the device can be revealed by calculating a so-called super-hologram, in which all measured particles of a large number of holograms are combined into one hologram. This technique is commonly used to select a subsection of the holography volume in which the detection efficiency is sufficiently uniform. However, it is impossible to translate these relative differences to an absolute measure of detection efficiency or precision and recall. With

our test holograms with CloudTarget in the sample volume, we can measure the absolute detection in precision and recall as a function of x,y,z-position and size of the particle.

## 585 5.2.1 Detection Dependency on z-Position

In Figure 6 we show precision and recall of the six individual TARGET-z holograms. Each hologram was recorded with CloudTarget with one photomask at a fixed z-position (TARGET datasets 2). We show recall and precision as a function of particle size for each z-position of the target assuming a Particle Classification Threshold of 0.3 (see section 3.2.2 and section 4.2).

In the top Figure 6 it can be seen that generally the recall increases with increased particle size, meaning smaller particles are harder to detect. For z-positions <10 cm we see a linear increase from 0 to 85% going from 4 to 7 µm in particle diameter. For larger particles the recall is not diameter dependent anymore and fluctuates around 90-95%. Particles further away, i.e. z >10 cm the recall is size dependent for all relevant particle diameters and is below 85% for most of the particle sizes and even below 60% for particles <10 µm.

In the lower panel of Figure 6, we see that precision is not dependent on the z-position in the volume. Precision is size-dependent for small measured particles  $<12 \mu m$ , where precision is worse for smaller particles. For particles  $>12 \mu m$  the precision fluctuates mostly around 80-100%.

We assume the measured recall to be realistic and do not see a source of inaccuracy. The complete knowledge about the FN (see Figure 3 FN = FN1 + FN2) makes the recall measured with CloudTarget more reliable than the recall when comparing to a manually annotated test dataset. As mentioned, the precision is likely to be influenced by ghost particles caused by reflection of the photomasks and therefore specific to CloudTarget holograms (see section 3.2.2, ghost particles in parallel plane are removed but presence of more ghost particles is likely). We can therefore assume the precision to be higher in in-situ holograms and the values shown here can be seen as a lower bound. A a more realistic assessment of precision is given by the test against manual annotations (Fig. 4) as FP (as oppsed to FN) are fully captured by the test of the classifier alone.

# 5.2.2 Detection Dependency on x-y-Position




As mentioned, the detection efficiency, i.e. the recall does not only depend on the distance to the camera (z-position) but also the x-y-position. For particles closer to the edge of the hologram cross section, meaning x-y closer to the camera edge, parts of the diffraction patterns are not recorded. This can be assumed to be a symmetric phenomenon and the effect on detection depends on z-position and particle size due to the difference in localisation and signal strength of diffraction pattern (see discussion in section 2 and Fig. 1 **Acquisition** schematic). Therefore, information about the particle near the edge is lost and detection is harder. To analyse this, we show the recall of particles in a hologram close (z = 9.9cm) and far away (z = 19.2cm) from the camera for different sizes of particles as a function of how much we exclude from the edges of the sample volume cross-section in Figure 7. We only exclude the edges in the last processing step, cutting them away from the effective sample volume. In

Figure 6. Top: Recall as a function of ground truth particle diameter. Bottom: Precision as a function of measured particle diameter. Shades of blue show results for different z-distances. Results are shown for PCT = 0.3 and 1 mm of the edges in x-y cut off. While recall strongly depends on z-distance, precision does not. Both recall and precision increase with particle size.

reconstruction, of course, the whole hologram is still used. This way none of the particles in the new effective sample volume are closer to the edge of the sensor than the chosen cutoff.

As expected, the general trend is the more we reduce the cross-section of the sample volume, the higher is the recall. The effect is in particular significant for small particles. For the hologram with particles closer to the camera sensor (low z) the recall saturates after excluding 1 or 2 mm from the edges. Detection of small particles ( $<15 \mu m$ ) further away in z, however still significantly improves when cutting off up to 5 mm from all edges which reduces the sample cross section from 1.5 cm x 1.5 cm to 0.5 cm x 0.5 cm. This can be explained by the larger spread in x-y of diffraction patterns for particles at large z. For large particles  $>20 \mu m$ , however, we see a saturation of recall when cutting 2 mm, a further reduction of cross section does not increase detection.


Figure 7. Influence of particle in x-y-position and size on recall. A shows schematically how the sample volume is limited. B shows how detection (recall) is improved for particles at z = 9.9 cm by limiting the sample volume in x-y-direction. In C it can be seen that detection improves significantly for particles further away from the camera z = 19.2 cm by excluding the edges of the sample volume. Detection towards the edges of the x-y plane is harder, especially at high z and for smaller particles. Therefore recall is increased by cutting the edges from the sample volume. The effect is stronger for small particles and at high z.

# 5.3 Droplet Sizing Accuracy




For final sizing, an object specific threshold (s-threshold, Figure 1) is applied to the in-focus amplitude crop and the areaequivalent diameter is determined. As described in section 2.2, we calculate the s-threshold using the method described in Schlenczek (2018) chapter 4.1.3 with an adaptable factor called *stretch Factor*, sF. The stretch Factor allows adjustment of the values of background level BL and particle level PL from which the threshold is then calculated as described in Schlenczek (2018). Other thresholding methods, such as the IsoData algorithm, did not prove to be successful.

CloudTarget holograms recorded in the laboratory can not be directly used to choose or validate the typical threshold-based sizing-methods for use on the in-situ holograms. We found that adjusting the s-threshold finding algorithm to match the Cloud-Target ground truth size (sF = 0.1) leads to significant under-sizing in the in-situ holograms as indicated by the inverse method via sign-matched filtering by Lu et al. (Lu et al., 2012) (to be discussed in more detail below) and a test with NIST glass beads (see 3.1.2). This can be associated to different signal to noise ratios in CloudTarget vs. in-situ holograms. A threshold algorithm that selects the correct threshold for accurate sizing in the cleaner CloudTarget holograms determines a threshold that is too strict in in-situ holograms due to darker background signal. Therefore it misses pixels of particles and under-sizes them.

The inverse method is threshold-independent and instead tries to match the diffraction pattern to determine size of a particle. In Lu et al. (2012) the inverse method for improved sizing is only experimentally tested on an approximately 50 µm sized droplet and its size-dependent performance is unknown. Using CloudTarget, the performance of the inverse method can now be tested for a wider range of particle sizes. Since the sizing ground truth provided by the TARGET holograms can not be

used to validate our threshold-based sizing-method, we chose to use the inverse method as an independent sizing reference. We applied the inverse method to 7000 predicted particles from the two flights. This dataset is called CLOUD-inv (see Table 2). The results of the CloudTarget test of the inverse method are shown in Figure 8, where the inverse method diameter  $d_{inv}$  is 645 shown against the ground truth diameter  $d_{at}$  for subsets of TARGET8-17 (target datasets that are within the z-range of reconstruction of in-situ holograms). The inverse method reliably sizes particles larger than  $d_{inv} > 12$  µm. If the size determined by the inverse method  $d_{inv}$  is smaller, it is not meaningful as can be seen in the larger scatter below the light blue line in Figure 8 A. The extreme outliers throughout all size ranges can be explained with particles being close to each other and the inverse method matching the diffraction pattern of a neighbouring particle. Testing our s-threshold algorithm against the 650 inverse method is therefore a valid method to test sizing for droplets  $d_{inv} > 12 \,\mu\text{m}$ , which can be used as a precious tool to fine tune sF for accurate sizing of in-situ holograms, for which we do not have a ground truth available. This is shown in bottom panel of Figure 8 for two different sets of in-situ holograms obtained in two different flights. We see that the optimal stretch Factor may vary slightly between the two flights but overall sF = 1.8 seems a good choice for minimizing the bias in the sizing. The comparison to the inverse methods for particles  $d_{inv} < 12 \, \mu \text{m}$  can be ignored as the inverse method does 655 not provide a reliable reference here as proven by the CloudTarget test. Still, we believe that adjusting the stretch Factor so that s-threshold particle sizes match the ones found by the inverse method for larger droplets also improves sizing of smaller droplets. Generally, there seems to be a constant trend in over- or under-sizing almost independent of particle size, depending on the sF that is corrected by adjustment of the stretch Factor. The correction is however smaller for particles <8  $\mu$ m. This effect is shown in Figure 8 B and C, where we correct the stretch Factor from what is best used in in-situ holograms (sF = 1.8) 660 so that it matches CloudTarget results (sF = 0.1) for particles >12  $\mu$ m. Admittedly, small particles are still oversized with CloudTarget-optimised sF = 0.1 but sizing error still is reduced.

Overall, CloudTarget helped confirming the accuracy of the inverse method. With the inverse method, we were able to determine the optimal stretch Factor for finding the s-threshold and size the droplets recorded in the in-situ holograms. We know that the stretch Factor is optimal for droplets >12  $\mu$ m and CloudTarget tests with different stretch Factors suggest that it is therefore also an improvement for smaller droplets. Comparing the s-threshold sizing to the inverse method does not allow a direct assessment of the distribution of absolute sizing errors of the s-threshold as the inverse method has an inherent scatter. Assuming, however, that the difference in sizing between CloudTarget holograms and in-situ holograms is a constant offset, the error distribution of sizing in in-situ holograms with sF=1.8 is the same as for CloudTarget holograms with sF=0.1. This indicates a standard deviation of the sizing error of about 2  $\mu$ m. Since absolute sizing error and error distribution width is largely constant along droplet sizes, the relative error in sizing decreases with droplet size.

# 5.4 Droplet Position Accuracy



In this section, we explore the capabilities of CloudTarget to verify the precision of 3D-position measurement in holography. The section is divided into two parts, where we examine the z- and x-y-position accuracy separately.

Figure 8. Determining the optimal stretch Factor sF for finding the sizing s-threshold. A: inverse sizing method (Lu et al., 2012) tested on TARGET holograms. B: s-threshold sizing method tested on TARGET holograms. C: sizing error of s-threshold with different stretch Factors. On TARGET holograms sF = 1.8 leads to significant over-sizing of about 2-3  $\mu$ m. The standard deviation is shown as dotted line, it is about 2  $\mu$ m independent of particle size or stretch Factor. D,E: sizing error with the inverse sizing method as the reference to determine optimal stretch Factor for particles >12  $\mu$ m where the inverse sizing method can be trusted.

# 675 5.4.1 z-Position Accuracy





The z-position of the particles is determined by identifying the focus plane. In principle, the focus plane is the plane where the particle appears in-focus with a sharp edge (see section 2). While the resolution of z-position is influenced by the  $\Delta z$  in the reconstruction, and with the  $\Delta z = 100$  µm chosen here, an average deviation of half the step size  $z_m - z_{gt} = 50$  µm) could be expected even with a near-perfect focus finder and precise z-position measurement.

The distance from image plane to photomask and hence the expected z of CloudTarget can be physically measured, but the accuracy is on the order of millimeters and hence not be usable as  $z_{gt}$ . But due to the use of the photomasks in CloudTarget, we know, that all particles are perfectly placed on a 2D plane within the volume. This plane is not necessarily perfectly perpendicular to the image plane but can exhibit a tilt in x-z and/or y-z.

We cannot measure the *absolute* error in z-position but determine how much the z-positions scatter around a perfect 2D plane. Since there is no reason to believe the z-position would have a bias towards over- or underestimation of z, we argue this scatter is a valid measure of accuracy of z-position. For each TARGET dataset we fit a plane in x-y-z-space to the positions of TPs as shown in Figure 9 **B** with  $z(x,y) = a_0 + a_1x + a_2y$ , which takes the tilt of the photomask into account. The offset in z from the measured  $z_m$  to the ideal  $z_{fit}$  is shown in Figure 9 **C** as a boxplot, which is a reasonable proxy measure for z-error for the different TARGET datasets each representing a different mean z-position. The first observation is that the uncertainty in z position of particles increases with z-position. For particles with low z 

Figure 9. Relative z-position accuracy is estimated by fitting a 2D plane through TP found in each TARGET dataset (B). The deviation in z from the measured particle position to the fitted 2D plane  $z_m - z_{fit}$  can be interpreted as random error in z. z-position error increases with the distance (C) from the image plane z but has no particle diameter dependence (D,E).

The dilation can be quantified by calculating the inter-particle distances  $s_{i,j} = \sqrt{(x_j - x_i)^2 + (y_j - y_i)^2}$  for all particle pairs in both the measured data of a hologram and the ground truth. This is an indirect measure of relative particle position without having to rely on non-exact matching through translation and rotation. Since holography gives us a 3D position for the particles we can correct the measured inter-particle distances for tilt in x-z and y-z. Since the misalignment seems to be mostly dependent on the photomask used and not so much on z-distance or hologram, we assume there is a deviation between expected ground truth position and actual ground truth position on the target. In fact, some of the photomasks purchased later showed only very slight positional deviations compared to the reconstructed holograms, suggesting that the deviations observed in some other photomasks were most likely due to manufacturing imperfections. To investigate this more thoroughly, we imaged two parts of the whole target with a Keyence vk-x200 3D laser microscope shown in red in Fig. 10 A as described in section 4.1. From these microscopy images we also extracted the inter-particle distances. The relative inter-particle distance differences to the expected ground truth  $\Delta s_{rel} = \frac{s_m - s_{gt}}{s_{gt}}$  are shown in Figure 10 E for the two microscopy images and three holograms. For each dataset the two boxes correspond to particle pairs from two different overlapping regions. Of course, only inter-particle distances of particles that appear in all measurements are considered. We see that microscopy agrees more with the distances measured from the holographic data than the theoretical ground truth data. We can therefore conclude that the  $\sim -1\%$  dilation

Figure 10. When x-y-positions of particles from measurement are overlayed they match perfectly in the center close to the tracking pattern ( $\mathbf{B}$ , particles not to scale). From  $\mathbf{C}$  and  $\mathbf{D}$  it can be seen that towards the edges the measured x-y-positions are typically smaller, the measured particles are placed further inward than the ground truth.  $\mathbf{E}$ : We show the relative difference of measured to expected inter-particle distances  $\Delta s_{rel}$  for different measurements of the target. We show the results for 3 holograms that tend to underestimate inter-particle distances but a reference microscope measurement also validates this. (left: particles in MICRO1, right: particles in MICRO2)  $\mathbf{F}$ : Absolute difference of measured to expected inter-particle distances  $\Delta s_{abs}$  as a function of expected inter-particle distance. A clear linear trend can be seen which can be interpreted as a constant dilation factor.  $\mathbf{G}$ :  $\Delta s_{abs}$  holography compared to expected ground truth and compared to the microscope images corrected with constant stretch factor which leaves random error which is on the order of 10  $\mu$ m.

of holography compared to the expected ground truth is an upper bound to the x-y-position error.


In Figure 10**F** we show the absolute difference of the measured inter-particle distances and the ground truth inter-particle distances  $\Delta s_{abs} = s_m - s_{gt}$  as a function of the ground truth inter-particle distances  $s_{gt}$  as an example for the hologram with photomask at z=5 cm. The linear relationship confirms the negative dilation of particle positions suspected from Fig. 10 **A-D**. Assuming the microscopy is accurate the holography over-sizes distances (i.e. positive dilation). This dilation is z-dependent 0.3% for z=5 cm 0.4% for z=9.9 cm and 0.8% for z=19.2 cm. This linear dependence could be easily explained by a not perfectly collimated diverging beam with an angle of less than  $0.02^{\circ}$ . If we, however, assume the ground truth particle positions to be correct we could not find a physical interpretation for the results .

As a lower bound for x-y-position error we corrected the results for the holograms for dilation compared to the expected ground truth of the photomask. For that, we calculate a dilation factor through a linear fit of  $\Delta s_{abs}(s_{qt})$  (as indicated in Fig. 10 **F**) and

correct the measured inter-particle data  $s_m$  with that factor. When we calculate the difference in inter-particle distances then, it can be interpreted as the random error (systematic error due to collimation removed, random error could be caused by errors in finding particle center) in distances  $\Delta s_{corr}$  shown in Figure 10 G. This random error is typically lower than 10  $\mu$ m but can be up to 40  $\mu$ m for large z.

Overall, CloudTarget in its current form is not able to give a precise answer to the question of position accuracy due to likely a small but significant dilation of 1% in position of the printed circles. We can, however, estimate a lower bound of  $10 \mu m$  error in x-y and an additional deviation of around 1-2% in distance measurements as an upper bound. For quantities, such as concentration or Liquid Water Content LWC measurements in clouds, the change in volume due to a dilated image with 1-2% is negligible. The accuracy of inter-particle distance is particularly interesting for measures of particle clustering like the Radial Distribution Function (RDF) g(r+dr), however inaccuracies in z are typically still larger and therefore the bigger challenge. Still, the inability to exactly quantify the x-y-position error with the current version of CloudTarget is a disadvantage. For future versions, photomasks with even higher position accuracy would be required to measure position accuracy more directly. Nevertheless, CloudTarget already makes it possible to record holograms with the exact same particles and therefore the same set of inter-particle distances that can be measured. Even if the absolute ground truth for the positions or inter-particle distances is not precisely known, comparing holograms can reveal inconsistencies that may be attributed to collimation issues.

## 6 Conclusions







# 6.1 Verification Capabilities of CloudTarget

We believe strongly that CloudTarget is a necessary tool for the characterisation of holography, but we also recognise its limitations and provide a brief overview of its advantages and disadvantages here. The CloudTarget was developed primarily to measure absolute detection efficiency — and has successfully fulfilled this task. With CloudTarget, we were able to quantify the recall rate as a function of position along the camera axes and within the hologram cross-section. While we trust the recall rate assessment with CloudTarget, the presence of ghost particles affects our precision measurement and results in lower values than expected in in-situ holograms of droplets. We account for a distinct ghost particle layer located 1.5 mm from real particles and remove these artifacts from the analysis. However, ghost particles caused by reflections may still exist within the actual z-layer, making them difficult to exclude. Given the high accuracy of the classifier, confirmed through manual labeling, we suspect the true precision is higher than suggested by the TARGET datasets. Nevertheless, CloudTarget measurements provide a robust and conservative lower bound on precision.

CloudTarget also helps to accurately determine sizing errors as it enables the validation of the sizing reference from the inverse method. The size error combined with the detection efficiency gives an estimate of the uncertainty of quantities such as liquid water content. Another use was to estimate the uncertainty in position measurements, but it remains unclear to what extent we can trust the theoretical inter-particle distances of CloudTarget's photomasks. However, the ability to record holograms with the same set of particles, i.e. with the same inter-particle distances, allows us to compare the relative positions of the particles between the holograms. At the very least, we can estimate the random error or scatter of the particle position and reveal a di-

vergence or convergence of the laser beam in the holographic system that leads to a positive or negative dilation of the relative positions. With the matching techniques used here, the position and size errors are insignificant for the precision and recall assessment.

As with all test holograms, CloudTarget holograms will most likely look different from in-situ holograms in terms of noise. In future, we think it is useful to take test holograms right before and after in-situ holograms are collected, so that contaminated windows, optics and their exact position and alignment is captured and can be tested. The effect of contamination of windows for example, can also be tested artificially in laboratory tests. While artificially introduced noise in the lab can never fully match the noise of in-situ holograms, some noise sources remain statistically similar, e.g. the laser instabilities, camera shot noise, reflections and other optical artifacts. As a result, the background and noise removal techniques should not be calibrated with CloudTarget holograms but always with in-situ holograms. Other than that, all processing steps can be evaluated and optimised with CloudTarget.

CloudTarget's reliance on photomasks restricts particles to discrete z-positions, which is a clear disadvantage. It also means that, when matching particle concentrations, the typical 3D distance between particles is smaller than it would be in a real scenario. Additionally, achieving a large number of z-positions requires multiple photomasks, which reduces transmission and can introduce reflections, degrading hologram quality in a way that is not representative of actual conditions. Overall, despite the limitations mentioned above, CloudTarget remains an invaluable tool for the evaluation and optimisation of holographic instruments. By measuring absolute detection efficiency by quantifying recall rates, estimating size errors and assessing positional uncertainties - all in a single setup - it provides essential information that is otherwise difficult to obtain. Combined with its ease of use, CloudTarget proves to be a robust and highly effective choice for quantifying holographic measurements.

## **6.2** Verification Results




In summary, we have shown how to decide on training data for holographic object classification and which evaluation steps are needed to determine potential errors and inaccuracies of the particle data output of holograms. We have demonstrated that with current verification methods for holographic systems there is still a gap for absolute quantification of detection efficiency. It is crucial to not blindly trust the complex and error prone process of hologram analysis without proper verification. Despite non-optimal performance of CloudTarget due to position inaccuracies of the printed circles and the effects of reflected light on the photomasks this gap in verification method is filled with CloudTarget. We can conclude for the specific properties of the holographic setup used in this study: a 5120 × 5120 pixel imaging sensor, a 3 µm effective pixel size, and 355 nm illumination:

- The CNN developed in this study can classify "objects" correctly when fed with moderate amount of training data. It is important to include rare objects (e.g. large particles). Due to better generalisation the CNN can be used fully automated and classification output increased from 10<sup>2</sup> (supervised decision trees) to 10<sup>4</sup> holograms per day.
- A verification of the classifier with manually annotated test is valid to choose the right classifier and assess precision (>90%) but can not accurately determine detection efficiency in terms of recall.

- While the CNN classifier shows promising generalization from in-situ training data to the CloudTarget dataset, and also to other holographic setups as demonstrated by (Thiede et al., 2025), its performance should still be evaluated using established or new verification methods such as CloudTarget before applying it to datasets with significantly different noise characteristics or from different instruments.
- The CloudTarget proved invaluable in finding recall is >90% for particles >7 um within (at least) z < 10 cm.
- To achieve good detection especially for small particles CloudTarget results suggests that is best to exclude  $\sim 2$  mm of the x-y-edges from the analysis.
- By combining the inverse method (Lu et al., 2012) with CloudTarget results the sizing threshold was optimised to not
   have a bias towards over- or under-sizing. The comparison suggests a standard deviation of sizing error of a about 2 μm
  - The focus position offset can be estimated with CloudTarget and we found the rms is below 150  $\mu$ m for particles with z < 10 cm while the step size of z-position  $\Delta z = 100$   $\mu$ m.
  - x-y-position analysis with CloudTarget revealed a potential diverging beam (<0.02 deg) in the holographic system that leads to slight stretching of inter-particle distances with increasing z.

# 810 Appendix A: CNN Layers



```
imageInputLayer([30 30 2], 'Normalization', 'zerocenter')
convolution2dLayer(10, 40, 'Stride', [1 1], 'Padding', [0 0 0 0])
batchNormalizationLayer
reluLayer
fullyConnectedLayer(2)
softmaxLayer
classificationLayer
```

# **Appendix B: CNN Training Options**


Momentum: 0.9000 InitialLearnRate: 1.0000e-03 LearnRateSchedule: 'none' LearnRateDropFactor: 0.1000 LearnRateDropPeriod: 10 1.0000e-04 L2Regularization: GradientThresholdMethod: 'l2norm' GradientThreshold: Inf MaxEpochs: 30 MiniBatchSize: 128 0 Verbose: VerboseFrequency: 50 ValidationData: П ValidationFrequency: 50 ValidationPatience: Inf Shuffle: 'once' CheckpointPath: ExecutionEnvironment: 'auto' WorkerLoad: П OutputFcn: П Plots: 'training-progress' SequenceLength: 'longest' 0 SequencePaddingValue: SequencePaddingDirection: 'right' DispatchInBackground: 0 ResetInputNormalization: 1 BatchNormalizationStatistics: 'population'

Author contributions. BT,KS and GB conceptualized CloudTarget verification. BT, OS, AE and GB developed the CNN classifier. BT, KS, OS and GB performed CloudTarget experiments and analysis. BT, OS, KS, AE, EB and GB interpreted the results. BT and GB wrote the initial draft of the paper. BT, OS, KS, AE, EB and GB proofread and edited the paper.

Competing interests. The authors declare no competing interests.

Code availability. Details on the CNN layers and structure used in this study can be found in Appendices A and B

Acknowledgements. We thank the Max Planck Institute for Dynamics and Self-Organization machine shop for building CloudTarget. Many thanks to Ayush Paliwal for helpful discussions about classification evaluation.

This work was partly supported by the Fraunhofer–Max Planck cooperation program through the TWISTER project. Birte Thiede was financially supported by a fellowship of the IMPRS for Physics of Biological and Complex Systems.

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
