# Peer review of "In-Line Holographic Droplet Imaging: Accelerated Classification with Convolutional Neural Networks and Quantitative Experimental Validation"

_EGUsphere, 2025_

## Referee Comment (RC1)

**Review of "In-Line Holographic Droplet Imaging: Accelerated Classification with Convolutional Neural Networks and Quantitative Experimental Validation" by Thiede et. al**

This manuscript presents CloudTarget, a calibration tool for holographic droplet images, and a CCN-based classification system for distinguishing between particles and artifacts when reconstructing holograms for cloud droplet studies. The authors demonstrate a method they call CloudTarget to quantify droplet detection efficiency, diameter measurement, and position accuracy using a series of chrome photomasks. The methodology is evaluated on several datasets and the performance of the method is evaluated quantitatively. The use of CCN is shown to achieve high accuracy in object classification while enabling faster data processing.

**General Comments**

This manuscript is very thorough in the reporting of results and it demonstrates high level of technical rigor. Using quantitative metrics and various datasets, the authors systematically compare the performance of their calibration method, CloudTarget, and the CCN used to classify particles and holograms. The paper is well written and organized, however, it can feel dense at some points. As it stands, the manuscript is detailed enough to serve as a good introduction to a beginner calibration methods for holograms of droplet. However, it may be worth considering trimming some content. Overall, I believe this is a useful manuscript to the scientific community and researchers using in-situ holographic data. Therefore, I recommend publication after minor revisions.

**Specific Comments**

1. Another source of optical artifacts comes from beam quality, which is a result of the quality of the light source, optics used to generate the beam, and any optics that may be used to project an image of the holographic field on a camera. The authors state that artifacts may come from dust and water droplets, however, it seems that the data is processed so that there is background subtraction, so I am unsure how these features might produce artifacts. Can the authors please clarify?

2. Related to the previous point, can the authors comment on why case 1 in Table 2 shows only 22 percent features are droplets? Presumably the other detected features are artifacts?

3. Can the authors discuss how hologram quality impacts the fine-tuning of CNN classifier? To the best of my understanding, the CNN needs to be trained for a specific hologram exposure and image quality. For example, if the beam quality or mean hologram exposure changes from one dateset to another, then the CNN

will need to be retrained on a dataset that has similar hologram image quality. Is this true? If this is the case, it would be useful if the authors acknowledge in a direct way that the CNN is highly sensitive to image quality. There is some discussion on this in the manuscript (Section 2.3), but it does not discuss changes in hologram quality.

4. The calibration targets used in the CloudTarget has particles which are deposited on a glass slide. Do the authors think the thickness of the glass slide (and subsequent change in index of refraction and optical path length) change their accuracy to measure droplet location in the $z$ direction?

5. In Section 5.2.2 a discussion on the diameter dependence of the droplet spatial position is discussed. This effect has been previously reported. For example, see https://doi.org/10.1088/1361-6501/ab79c6

6. The authors acknowledge that the small glass beads tend to clump together. How are these regions identified and measured? Are they presumed to be artifacts?

7. As mentioned in the general comments, parts of the paper seem dense. It may be worth considering trimming some content. For example, the Fraunhofer diffraction equations are standard in holography and well-covered in cited references. Another example is the discussion of alternative calibration methods can read more like a part of a review article.

**Technical Comments**

1. Line 4: consider rephrasing "... with a customized pattern of opaque circles as a verification tool" to something like "... with a customized pattern of opaque circles, serving as a verification tool"

2. Line 20: fix "on board of an aircrafts"

3. Line 122: remove comma "We discovered, that ..."

4. Line 465: there seems to be a grammar error in "from CloudTarget as a reference"

5. Line 604: grammar error in "... and don not see ..."

---

## Author Comment (AC1)

**Reply to Anonymous Referee #1**

**July 31,2025**

Dear Reviewer,

Thank you very much for reviewing our manuscript. We are very grateful for the extremely helpful and constructive comments. In the following, we provide point-by-point replies to the points raised in your report. We have marked the original text of the review in blue colour and our response in black colour.

*This manuscript presents CloudTarget, a calibration tool for holographic droplet images, and a CCN-based classification system for distinguishing between particles and artifacts when reconstructing holograms for cloud droplet studies. The authors demonstrate a method they call CloudTarget to quantify droplet detection efficiency, diameter measurement, and position accuracy using a series of chrome photomasks. The methodology is evaluated on several datasets and the performance of the method is evaluated quantitatively. The use of CCN is shown to achieve high accuracy in object classification while enabling faster data processing.*

*This manuscript is very thorough in the reporting of results and it demonstrates high level of technical rigor. Using quantitative metrics and various datasets, the authors systematically compare the performance of their calibration method, CloudTarget, and the CCN used to classify particles and holograms. The paper is well written and organized, however, it can feel dense at some points. As it stands, the manuscript is detailed enough to serve as a good introduction to a beginner calibration methods or holograms of droplet. However, it may be worth considering trimming some content. Overall, I believe this is a useful manuscript to the scientific community and researchers using in-situ holographic data. Therefore, I recommend publication after minor revisions.*

We appreciate the positive assessment of our manuscript, particularly the recognition of its technical rigor and its potential value to the community working with in-situ holographic data. We will address each of the comments in detail, providing point-by-point responses in which we indicate the changes made in the revised manuscript. The response to comment C7 acknowledges and adresses the observation that the manuscript can feel dense at times, mentioned by the reviewer in the general comment. We are thankful for the encouraging remarks and valuable suggestions.

> *(C1): Another source of optical artifacts comes from beam quality, which is a result of the quality of the light source, optics used to generate the beam, and any optics that may be used to project an image of the holographic field on a camera. The authors state that artifacts may come from dust and water droplets, however, it seems that the data is processed so that there is background subtraction, so I am unsure how these features might produce artifacts. Can the authors please clarify?*

(A1): We appreciate the reviewer raising this important point.
We have addressed noise sources in the original manuscript in Section 2.1, line 112, where we list potential systematic influences including laser beam inhomogeneities, contaminated optics, and mechanical vibrations. In the revised manuscript we have changed "contaminated optics" to "imperfect contaminated optics" to to highlight the fact that of course, as the reviewer correctly pointed out, the optics have an effect themselves, even if they were not contaminated.
While our processing pipeline includes static background removal using an advanced median division method based on neighboring holograms, this approach primarily removes persistent features. Noise that appears only in a single or a very small number of holograms, such as those caused by transient dust or water droplets on optics, or sudden beam inhomogeneities, can remain in the data and affect the reconstructions.

Additionally, in Section 2.2, line 157 of the original manuscript, we state that reconstructed planes may contain not only in-focus particles but also diffraction patterns from out-of-focus particles and residual background signals that were not fully removed which we believe is the main reason for our artifacts.

*(C2): Related to the previous point, can the authors comment on why case 1 in Table 2 shows only 22 percent features are droplets? Presumably the other detected features are artifacts?*

(A2): Yes, that is correct, the remaining features are classified as artifacts. The 22% figure in Table 2 refers to the proportion of detected objects that were manually annotated as true droplets in the training dataset. These "objects" that make up training or testing datasets correspond to dark regions identified in the holograms, which can be either real particles or various types of artifacts.
Our CNN-based classifier was trained not only to recognize the appearance of real particles, but also to learn the typical characteristics of noise and artifacts (classification into 2 classes "particle" and "artifact", see line 182 in the original manuscript: "The aim is to achieve a classification into the two classes particle and artifact and therefore the output of the CNN should be a likelihood of the object being in each class."). By including artifacts in the training data, it should improve the accuracy of distinguishing droplets from background or spurious signals in holographic reconstructions.
We have slighlty reworded the caption of Table 2 in the revised manuscript to further clarify training and testing datasets include both particles and artifacts.

*(C3): Can the authors discuss how hologram quality impacts the fine-tuning of CNN classifier? To the best of my understanding, the CNN needs to be trained for a specific hologram exposure and image quality. For example, if the beam quality or mean hologram exposure changes from one dateset to another, then the CNN 1will need to be retrained on a dataset that has similar hologram image quality. Is this true? If this is the case, it would be useful if the authors acknowledge in a direct way that the CNN is highly sensitive to image quality. There is some discussion on this in the manuscript (Section 2.3), but it does not discuss changes in hologram quality.*

(A3): We thank the reviewer for raising this important point. Variations in hologram quality, such as differences in beam uniformity or exposure, can influence the overall signal-to-noise ratio and affect the quality of object detection.
However, the input to the CNN consists of cropped image patches from the reconstruction, which is performed on the filtered and more importantly normalised hologram (normalisation was mentioned in line 134 of the original manuscript). This normalisation helps to reduce the sensitivity of the CNN to absolute intensity differences between datasets/ holograms. In the revised manuscript, we added this information about normalization additionally in line 178, where the CNN classifier input is described, to highlight this point further ("One image is the amplitude and the other the phase, both from the focus planes of the objects determined in the reconstruction of the normalized hologram.").
In our experiments, we observed that the CNN classifier generalizes well across datasets with differing hologram properties. For example, as described in the paper, although the network was trained only on in-situ flight data, it performs well on the laboratory CloudTarget dataset. It also shows good performance on a dataset from the HoloTrack instrument, which includes larger droplets and a different optical setup (see https://doi.org/10.5194/egusphere-2025-1774). Nevertheless, we agree that when applying the classifier to new datasets with substantially different imaging conditions or instrument configurations, it is important to validate and, if needed, improve its performance by finetuning the network with additional training data. We

have updated the manuscript in Section 6.2 and added a bullet point to more clearly emphasize this recommendation. ("While the CNN classifier shows promising generalization from in-situ training data to the CloudTarget dataset, and also to other holographic setups as demonstrated by Thiede et al. 2025, its performance should still be evaluated using established or new verification methods such as CloudTarget before applying it to datasets with significantly different noise characteristics or from different instruments.")

*(C4): The calibration targets used in the CloudTarget has particles which are deposited on a glass slide. Do the authors think the thickness of the glass slide (and subsequent change in index of refraction and optical path length) change their accuracy to measure droplet location in the z direction?*

(A4): Yes, the presence of any glass surface, such as protective windows in the holographic setup or the glass substrate of the calibration target, can affect the optical path length and thus the measured absolute z-position due to the change in refractive index.
In our in-situ cloud applications, we are not concerned with the absolute z-position but rather with relative positioning accuracy and therefore do not correct the effect on optical path length of e.g. protective windows. In the CloudTarget experiments presented in the manuscript, only one photomask was used at a time. The glass of the CloudTarget has a thickness of 2.3 mm and is made of quartz (n = 1.46). This would lead to a shift of approximately 1 mm in optical path length if not accounted for, which even for our smallest z-distance of 5cm is only 2%.
The absolute z-distances provided throughout the manuscript are primarily used to illustrate trends with depth. In this context, the potential 1 mm optical path deviation is small relative to the measurement uncertainties and does not impact our conclusions. To clarify we have added "Therefore, in the tests described in this paper only a single photomask was used in CloudTarget, which was altered in z-position. The z-positions of the photomask is given as the mean reconstructed distance of the identified particles from the image plane throughout section 5. The change in optical path length due to the refractive index of n = 1.46 of the 2.3 mm thick photomask is approximately 1 mm and neglected here." in line 321 of the revised manuscript. Further, our evaluation on z-position accurcay focuses on relative z-positioning rather than absolute z-measurements. As stated in the original manuscript (line 690), *"We cannot measure the absolute error in z-position but determine how much the z-positions scatter around a perfect 2D plane. Since there is no reason to believe the z-position would have a bias towards over- or underestimation of z, we argue this scatter is a valid measure of accuracy of z-position."* Section 5.4.1 therefore is unaffected by this issue.

*(C5): In Section 5.2.2 a discussion on the diameter dependence of the droplet spatial position is discussed. This effect has been previously reported. For example, see https://doi.org/10.1088/1361-6501/ab79c6*

(A5): We thank the reviewer for the reference. While the cited work addresses diameter-dependent spatial effects, to our knwoledge it does so in the context of a different imaging modality (not digital in-line holography). In our Section 5.2.2, we specifically discuss and quantitatively assess, how detection challenges arise at the edges of the cross-sectional volume in holographic reconstructions, where parts of a droplet's diffraction pattern may fall outside the camera's field of view. This leads to reduced detection efficiency for smaller droplets near the volume boundaries. As such, the underlying mechanism and imaging constraints differ from those discussed in the referenced study.

*(C6): The authors acknowledge that the small glass beads tend to clump together. How are these regions identified and measured? Are they presumed to be artifacts?*

(A6): We thank the reviewer for this observation. The small glass beads were primarily used during early validation tests and were not included in the main results presented in the manuscript. For this reason, we did not include figures or detailed discussions about them, they are only briefly mentioned in Section 3.1.2 as part of the established validation approaches we discussed. In our preliminary tests, we observed that the CNN did classify some clumps of beads as particles.

We have now made that clearer by adding "As at least parts of the clusters are classified as particles in our processing, this makes it increasingly challenging to identify the measured sizes corresponding to individual beads instead of bead clusters." In line 268 of the revised manuscript. We did not quantify exactly what fraction of the clumps were identified as particles by the CNN.

When measuring monodisperse glass beads, the measured size distribution was clearly multimodal with the first mode corresponding to the bead diameter, and larger modes suggesting the presence of bead aggregates. Manual inspection of the reconstructions for identified particles in the larger modes confirmed that these were often clumps of multiple beads. Although the first peak in the observed distribution aligned reasonably well with the actual bead diameter, the modes were not clearly enough separated which limited our ability further to use these data for fine-tuning the stretch factor in the sizing method.

We want to emphasize however, that the bead aggregates might be less of a problem if the injection method is improved (as pointed out by Reviewer 2, comment 3, and now mentioned in the revised manuscript in line 271) but is not the only limitation of the glass bead method. More important is the inability to compare particle measurements one-to-one to a ground truth as highlighted now more clearly in line 264 of the revised manuscript.

*(C7): As mentioned in the general comments, parts of the paper seem dense. It may be worth considering trimming some content. For example, the Fraunhofer diffraction equations are standard in holography and well-covered in cited references. Another example is the discussion of alternative calibration methods can read more like a part of a review article.*

(A7): We appreciate the reviewer's comment and fully understand the concern regarding the manuscript's density. We are aware that the paper is relatively long, however made an effort to keep it concise while maintaining clarity and completeness. We believe the Fraunhofer equation in particular allows insights into why size-, z- and xy- dependent detection efficiency is to be expected. Similarly, we felt that a more detailed discussion of alternative calibration approaches was important to clearly show the novelty and motivation behind *CloudTarget*, and to help contextualize our contribution in relation to existing methods.

Overall, we had a detailed look again into whether the paper could be significantly shortened in the mentioned areas and have removed the detailed describtion of the reconstructions (lines 134 to 149 in original manuscript) as this is described in detail in Fugal 2009 and a citation seems sufficient. While other steps of our processing chain are also in parts standard procedures, we believe a complete overview of the steps is necessary for reproducibility and transparency.

*(C8): Technical Comments*
*1. Line 4: consider rephrasing "... with a customized pattern of opaque circles as a verification tool" to something like "... with a customized pattern of opaque circles, serving as a verification tool"*
*2. Line 20: fix "on board of an aircrafts"*
*3. Line 122: remove comma "We discovered, that ..."*
*4. Line 465: there seems to be a grammar error in "from CloudTarget as a reference"*
*5. Line 604: grammar error in "... and don not see ..."*

(A8): We thank the reviewer for pointing the technichal errors out. The issues have been corrected in the revised manuscript.

We provide an additional version of the revised manuscript in which all changes are clearly marked, including those made in response to comments from the other reviewers.

Additionally, lines 141–149, 575, and 597–600 were removed, as they referred to a low-pass filter applied during reconstruction that is not used in our process and was incorrectly mentioned. This has now been clarified in line 137 of the revised manuscript, where Fugal (2009) is cited to describe the reconstruction process and explicitly stated that the method is applied without frequency low-pass filtering. The reference Paliwal 2025 in the list of references was corrected.

---

## Author Comment (AC2)

Reply to Anonymous Referee #2

July 31,2025

Dear Reviewer,

Thank you very much for reviewing our manuscript. We are very grateful for the extremely helpful and constructive comments. In the following, we provide point-by-point replies to the points raised in your report. We have marked the original text of the review in blue colour and our response in black colour.

*This is good work addressing a practical problem for holographic cloud probes. The results are important for the atmospheric science community and I would expect that this paper will be published fairly easily after some minor revisions, mainly identified already by the other reviewer.*

We thank the reviewer for their positive assessment of our work and for recognizing its relevance to the atmospheric science community.

*(C1): The authors use the standard Huygens-Fresnel reconstruction method, which is indeed appropriate here. I do wonder though if they have considered the more accurate angular spectrum method, and if so, why they chose not to use it instead. Probably there is not much difference for such large particles.*

(A1): We thank the reviewer for this comment. We would like to clarify that there may have been some misunderstanding due to how the reconstruction method was presented in the manuscript. While the Huygens–Fresnel diffraction was shown in integral (spatial-domain) form for illustrative purposes, we are in fact using a frequency-domain implementation (see line 140 in the original manuscript: "The reconstruction is implemented in Fourier space with a Huygens–Fresnel kernel in filtering form as explained in Fugal et al. (2009)"). To our understanding, this approach is equivalent to what is also referred to as the Angular Spectrum Method. To reduce confusion and respond to the other reviewers feedback about clarity and density, we have since removed the explicit spatial-domain integral form from the revised manuscript (Cemoved lines 134-149 of original manuscript) and clarified in the text that the frequency-domain method described by Fugal (2009) is the one used for wavefield propagation.
It is also worth noting that the focus of this work was not on modifying or optimizing the reconstruction process itself, but rather on the subsequent classification and evaluation. The reconstruction follows the method described in Fugal (2009) (now corrected in the revised manuscript to indicate that no low-pass filtering is applied), with only minor adjustments to background filtering. Our main contributions lie in the downstream processing including classification and evaluation methodology.
We hope this clarification resolves the concern.

*(C2): Around line 55, the authors address the promising concepts of skipping reconstruction. Because the particles here are spheres, the problem is actually simplified quite a bit. I wonder if the authors are aware of the very fast, very simple method in Denis et al. to size spherical particles quite accurately without reconstruction? See: Denis, Loïc, Corinne Fournier, Thierry Fournel, Christophe Ducottet, and Dominique Jeulin. "Direct extraction of the mean particle size from a digital hologram." Applied Optics 45, no. 5 (2006): 944-952.*

(A2): We thank the reviewer for pointing us to this very relevant and interesting reference. We were not previously aware of the method by Denis et al., and we agree that it offers a promising approach for quickly estimating the mean particle size without reconstruction, particularly in the case of spherical particles. We have added a citation to this work in the discussion of new

approaches to skip reconstruction in line 52 of the revised manuscript. The method, however, focuses on extracting only the mean particle size. The main adavantage of in-situ holography in clouds is the resolution of individual droplets and their three-dimensional spatial position and therefore the suggested method can not fully replace the reconstruction procedure. We do see value in potentially implementing this approach as a quick first-look tool for our datasets and greatly appreciate the suggestion.

*(C3): Regarding clumping small glass beads around line 275, there is an effective method to deal with this effect. Simply use sonification as shown in Fig. 6 in Giri, Ramesh, and Matthew J. Berg. "The color of aerosol particles." Scientific Reports 13, no. 1 (2023): 1594.*

(A3): We thank the reviewer for highlighting this helpful method. We have added a reference to the suggested paper in the revised manuscript (line 271 "The use of ultrasonic dispersion techniques, such as those described in \citet{giri2023color}, may improve the separation of beads and enhance the method's applicability in this regime.") and agree that using ultrasonic sonication can be effective in reducing bead clumping. However, even with improved dispersion, the glass-bead approach remains only suited for comparing size distribution and does not allow for a direct one-to-one comparison between known and measured diameters for individual droplets. As we argue in the paper (highlighted better in revised manuscript line 264), precise validation of individual droplet sizes is critical, which still limits the applicability of calibration beads.

We provide an additional version of the revised manuscript in which all changes are clearly marked, including those made in response to comments from the other reviewers.
Additionally, lines 141–149, 575, and 597–600 were removed, as they referred to a low-pass filter applied during reconstruction that is not used in our process and was incorrectly mentioned. This has now been clarified in line 137 of the revised manuscript, where Fugal (2009) is cited to describe the reconstruction process and explicitly stated that the method is applied without frequency low-pass filtering. The reference Paliwal 2025 in the list of references was corrected.

---

## Author Comment (AC3)

**Reply to Anonymous Referee #3**

**July 31,2025**

Dear Reviewer,

Thank you very much for reviewing our manuscript. We are very grateful for the extremely helpful and constructive comments. In the following, we provide point-by-point replies to the points raised in your report. We have marked the original text of the review in blue colour and our response in black colour.

*The manuscript "In-Line Holographic Droplet Imaging: Accelerated Classification with Convolutional Neural Networks and Quantitative Experimental Validation" by Thiede et al shows a very interesting new technique to improve assessment of holographic instruments and improvement of data quality. I enjoyed reading this manuscript and suggest minor revisions.*

We appreciate the reviewer's positive and encouraging feedback and are glad to hear that the manuscript was enjoyable to read.

> *(C1): I believe the manuscript could be more concise in several areas when well-known topics are being discussed, even though they are presented nicely and thoroughly here. It might help to give focus on the actual new results.*

(A1): We thank the reviewer for the suggestion to improve conciseness, particularly in sections discussing well-established concepts. We were already mindful of the overall length when preparing the original manuscript and tried to include only what we considered necessary. We revisited the manuscript again and removed the detailed description of the reconstruction methods (lines 134 to 149 in original manuscript), as a citation of Fugal 2009, where it is described in great detail, seems indeed sufficient.

We believe, other sections that do not present new results, still provide important context and justification for our approach and have thus decided to keep them in the revised manuscript. This includes e.g. the overview of established verification method as this demonstrates the need for a tool like CloudTarget and the complete overview of our processing steps for transparancy and reproducibility. We hope the reviewer can see our efforts to improve conciseness and agrees that the remaining content adds value in terms of clarity and motivation.

> *(C2): In table 2 it is hard to see which line the notes belong to, maybe separating them would make it clearer.*

(A2): We have added additional vertical spacing to separate the notes more clearly.

> *(C3): in line 322 you say 3 or 4 - when is it which? under which circumstances is it which?*

(A3): We have changed this to three as we start to see a slight decrease in recall when using three photomasks, which becomes more significant when using four.

> *(C4): same in line 385: is it 1 or 2?*

(A4): The labeling procedure varied between datasets and was either conducted by one operator or two operators to enable inter-comparison. For each dataset the number of operators is indicated in Table 2 and we have made that clearer now by adding "(as indicated in Table 2)" to this line in the revised manuscript in line 384.

*(C5): typos:*
*Line 148 typo - -*
*Line 256 typo an*
*Line 399 typo: filtering*
*Line 621: space missing*

(A5): We thank the reviewer for pointing out the minor issues related to typos. These have been corrected in the revised manuscript.

We provide an additional version of the revised manuscript in which all changes are clearly marked, including those made in response to comments from the other reviewers.
Additionally, lines 141–149, 575, and 597–600 were removed, as they referred to a low-pass filter applied during reconstruction that is not used in our process and was incorrectly mentioned. This has now been clarified in line 137 of the revised manuscript, where Fugal (2009) is cited to describe the reconstruction process and explicitly stated that the method is applied without frequency low-pass filtering. The reference Paliwal 2025 in the list of references was corrected.

---

## Author Response (AR2)

Dear Dr. Lelli,

We thank you for your careful handling of our manuscript and for your positive decision regarding its acceptance. We also value your insightful request for clarification.

We understand the source of the potential confusion. The wavefront reconstruction in our work is indeed performed using the angular spectrum method. In the revised manuscript, we have now explicitly referenced Goodman (2005) rather than only citing the specific implementation by Fugal et al. (2009). Specifically, in line 135 we now state:

"Subsequently, the filtered hologram is normalized and then reconstructed by propagation of the wavefront along the z-axis with the angular spectrum method (see e.g. chapter 3.10 in Goodman, 2005). The Reconstruction via angular spectrum method, i.e. implemented in Fourier space, is performed with the a Huygen-Fresnel kernel in filtering form, as explained in Fugal et al. (2009) Eq. 2-5 without additional frequency filters."

Additionally, in some parts of the manuscript, we previously used the term "reconstruction" when we were actually referring to the combined process of wavefront reconstruction followed by object extraction (i.e., application of a global threshold and focus finder, as described starting in line 142 of the revised manuscript). To avoid ambiguity, we have now replaced these instances with "reconstruction and object extraction," reserving "reconstruction" alone for the standard angular spectrum method step.

We hope these revisions solve the issue. Additionally we have updated Figure 10 as there was a mistake in the labeling of the panels ("top left" and "bottom right"). We have uploaded the revised manuscript and another version that marks the new changes.

Best regards, Authors